

**Contribution of Surface Solar Radiation and Precipitation to Spatiotemporal**
**Patterns of Surface and Air Temperature Warming in China from 1960 to 2003**
Jizeng Du[1,2], Kaicun Wang[1,2*], Jiankai Wang[3], Qian Ma[1,2]
[1]College of Global Change and Earth System Science, Beijing Normal University,
Beijing, 100875, China
[2]Joint Center for Global Change Studies, Beijing 100875, China
[3]Chinese Meteorological Administration, Beijing, 100081, China
**Corresponding Author**: Kaicun Wang, College of Global Change and Earth System
Science, Beijing Normal University. Email: kcwang@bnu.edu.cn; Tel: +086 10-
58803143; Fax: +086 10-58800059.



**Abstract**
Although the global warming has been successfully attributed to the elevated
atmospheric greenhouses gases, the reasons for spatiotemporal patterns the warming
rates are still under debate. In this paper, we report surface and air warming based on
observations collected at 1977 stations in China from 1960 to 2003. Our results show
that the warming of daily maximum surface ($T_{s\text{-}max}$) and air ($T_{a\text{-}max}$) temperatures
showed a significant spatial pattern, stronger in the northwest China and weaker in
South China and the North China Plain. These warming spatial patterns are attributed
to surface shortwave solar radiation (SSR) and precipitation, the key parameters of
surface energy budget. During the study period, SSR decreased by $-1.50$ W m$^{-2}$ 10yr$^{-}$
$^{1}$ in China and caused the trends of Ts-max and Ta-max decreased by 0.139 and 0.053 °C
10yr$^{-1}$, respectively. More importantly, South China and the North China Plain had an
extremely higher dimming rates than other regions. The spatial contrasts of trends of
$T_{s\text{-}max}$ and $T_{a\text{-}max}$ in China are significantly reduced after adjusting for the impact of SSR
and precipitation. For example, the difference in warming rates between North China
Plain and Loess Plateau reduce by 97.8% and 68.3% for Ts-max and Ta-max
respectively. After adjusting for the impact of SSR and precipitation, the seasonal
contrast of $T_{s\text{-}max}$ and $T_{a\text{-}max}$ decreased by 45.0% and 17.2%, and the daily contrast of
warming rates of surface and air temperature decreased by 33.0% and 29.1% over China.



This study shows an essential role of land energy budget in determining regional
warming.
**1. Introduction**

With the rapid development of observational data and the simulation abilities of

climate models, global warming has been regarded as undeniable (Hartmann et al.,
2013). The increase in anthropogenic greenhouse gases and other anthropogenic
impacts are believed to be the primary cause of global warming. However, there are
significant spatial and temporal heterogeneities in climate warming, i.e., faster warming
rates in semiarid regions and a "warming hole" in the central United States (Boyles and
Raman, 2003; Huang et al., 2012), which represents a major barrier to the reliable
detection and attribution of global warming (Tebaldi et al., 2005; Mahlstein and Knutti,
2010). Furthermore, the uncertainties in model simulations generally increase from the
global to the regional scale because of uncertainty in regional climatic responses to
global change (Hingray et al., 2007; Mariotti et al., 2011). Therefore, it is crucial to
research not only the spatial and temporal patterns of regional climate changes but also
regional climatic response mechanisms to global change. This approach can improve
confidence in the detection and attribution of global climate change and prediction of
future regional climate change.





The spatial heterogeneity of climate warming can be attributed to local climate

factors and anthropogenic factors (Karl et al., 1991). For the former, local determining

factors such as cloud amounts and precipitation can significantly influence regional

warming speeds (Hegerl and Zwiers, 2007; Lauritsen and Rogers, 2012). Those spatial

heterogeneities in climate-factor trends make important contributions to various

changes in the land-surface energy balance. Existing studies have indicated that an

increase in clouds can diminish downward shortwave solar radiation to the land surface,

thus reducing the daytime temperature (Dai et al., 1997; Zhou et al., 2010; Taylor et al.,

2011) while potentially increasing nighttime temperatures by intercepting outgoing

longwave radiation (Shen et al., 2014; Campbell and VonderHaar, 1997).

Precipitation can alter the proportion of surface absorbed energy partitioned into

sensible heat flux and latent heat flux and therefore has an inevitable impact on both

land-surface and near-surface air temperatures (Wang and Dickinson, 2012; Wang and

Zhou, 2015). In addition, precipitation plays a key role in the soil thermal inertia and

surface vegetation, causing important feedback to regional and global warming (Wang

and Dickinson, 2012; Seneviratne et al., 2010; Ait-Mesbah et al., 2015; Shen et al.,

2015).

In addition to local climate factors, anthropogenic emissions of aerosols have a



significant effect on the regional climate system. Studies indicated that improving air
quality in recent decades has led to brightening over North America and Europe (Wild,
2012; Vautard et al., 2009), whereas surface shortwave solar radiation (SSR) has
declined in East Asia and India with increasing air pollution (Xia, 2010; Menon et al.,
2002; Wang et al., 2012; Wang et al., 2015a). Consequently, the variation in SSR may
have an impact on both local and global climate change (Wild et al., 2007; Wang and
Dickinson, 2013b).

Land cover change can also alter the energy exchange between the land surface

and the atmosphere; moreover, it has the potential to impact regional climate (Falge et
al., 2005; Bounoua et al., 1999; Zhou et al., 2004). Previous studies have suggested that
urbanization and other land-use changes contribute to promoting the warming effect
caused by greenhouse gases (Kalnay and Cai, 2003; Lim et al., 2005; Chen et al., 2015).
Overall, the impacts of these factors on climate change may be very important on the
regional scale, leading to a marked spatial difference in regional climate change,
whereas they are usually omitted from the detection and attribution of climate change
on the global scale (Karoly and Stott, 2006).

China has a vast territory and abundant types of climactic zones stretching from

tropic to cold temperate, with a special alpine climate over the Tibet Plateau. In addition,



dramatic economic development and explosive population growth in recent decades has
caused significant land cover change and serious air pollution, including frequent haze
events (Yin et al., 2016; Cheng et al., 2014; Wang et al., 2016). The climatic diversity
and intensive human activity in this region will likely lead to a unique response to global
warming with obvious spatial differences in climate change.

Karl et al. (1991) had analyzed the observational records for the period 1951-1989,

finding that China's temperature warming trends were faster than those of the United
States but slower than those of the former Soviet Union. Several studies had revealed
that the warming rate in Northwest China had been approximately 0.33-0.39 °C $10yr^{-1}$
during the second half of the last century (Li et al., 2012; Zhang et al., 2010), which
was significantly higher than the average warming rate over China (0.25 °C $10yr^{-1}$)
(Ren et al., 2005) or that on a global scale (0.13 °C $10yr^{-1}$) (Hegerl and Zwiers, 2007).
Air temperatures (Ta) over the Tibet Plateau have increased by 0.44 °C $10yr^{-1}$ over the
last 30 years (Duan and Xiao, 2015), which was considerably faster than the overall
warming rate in the Northern Hemisphere (0.23 °C $10yr^{-1}$) and worldwide (0.16 °C
$10yr^{-1}$) (Hartmann et al., 2013). Understanding the characteristics and mechanisms of
regional climate change is critical to advancing the knowledge and predication of future
climate change.





106   Ta is a common metric for judging climate change on the global or regional scales.

107  However, land surface temperature (Ts) is beginning to play an increasingly important

108  role in climate change research because it has the distinct advantage of being directly

109  related to the land surface energy budget. Previously, $T_s$ values used in regional climate

110  research are primarily derived from satellite retrievals or reanalysis datasets (Weng et

111  al., 2004; Peng et al., 2014), both of which have good global coverage but questionable

112  accuracy and integrity. Furthermore, satellite-derived $T_s$ values are only available under

113  clear sky conditions, limiting their application to climate change studies.

114   In China, $T_s$ has been measured as a conventional meteorological observation item

115  by nearly all weather stations, as is $T_a$. This study found that observations of $T_s$ have a

116  good relationship with $T_a$ in terms of spatial-temporal patterns and can equally

117  accurately reflect the characteristics of climate change. More importantly, $T_s$ is more

118  sensitive to the local land surface energy budget, particularly surface solar radiation

119  (SSR) and precipitation.

120   From the perspective of energy, both SSR and precipitation are key factors

121  controlling the land surface energy budget; therefore, their changes most likely cause

122  regional differences in the warming rate of Ta (Wild, 2012; Manara et al., 2015;

123  Hartmann et al., 1986). For the first time, this study analyzed the relationship between




SSR (and precipitation) and $T_a$ or $T_s$ in terms of their spatial–temporal patterns and
further quantified the impact of the variations of SSR and precipitation on $T_a$ and $T_s$ in
China for the period of 1960–2003.

This paper is organized as follows: Section 2 introduces the data and method used

in the study. Section 3 includes three parts: the first part describes the spatial and
temporal pattern of climate warming over China; the second part analyzes the impact
of the variation in SSR and precipitation on $T_a$ and $T_s$; and the third part illustrates the
spatial and temporal pattern of the warming trend of $T_a$ and $T_s$ after adjusting for the
impact of SSR and precipitation. The adjustment removed impact of land-atmosphere
interaction on the warming, leaving impact of large scale warming caused by the
elevated atmospheric greenhouse gases substantially. Our results show that adjustment
substantially reduced the spatial contrast of warming trends of Ta and Ts in China,
which is agree with the expectation of global warming. A summary and discussion are
presented in Section 4.
**2. Data and method**
**2.1. Data**

The meteorological observational data used in this study are recently released daily





meteorological datasets, including the China National Stations' Fundamental Elements
Datasets V3.0 (CNSFED V3.0), which can be downloaded from the China's National
Meteorological Information Center (http://data.cma.gov.cn/data) (Cao et al., 2016).
This dataset includes $T_s$, $T_a$, the barometric pressure, relative humidity, and sunshine
duration. All of the observational records of the climate variables include quality
control and homogenization of the processes of data acquisition and compilation.

Figure 1 shows that the number of stations used in this study (1977 selected

stations from a total of 2479 stations) is abundant and significantly greater than in
previous studies (i.e., 57-852 stations) (Kukla and Karl, 1993; Shen and Varis, 2001;
Liu et al., 2004; Li et al., 2015); therefore, the observational data have better spatial
coverage and higher confidence of detecting regional climate change (Fig. 1). Our study
is the first to use the observations of $T_s$ for research into regional climate change.

Observations of $T_s$ at weather stations are different from data retrieved via other

approaches, such as satellite data and reanalysis. All of the observational fields of $T_s$
are 4 m × 2 m square bare land plots in a weather station. The surface of the
observational field must be kept loose, grassless, flat, and at the same level as the
ground of the weather station. Three thermometers are placed on the surface of the
observational field, including a surface thermometer, a surface maximum thermometer,





and a surface minimum thermometer. The thermometers are deposited on the surface
of the observational field horizontally: half of each thermometer is embedded in the soil
and the other half is exposed to the air. When the observational field is covered by snow,
the thermometers are removed from the snow and placed on the snow surface. In
addition, the exposed parts of the thermometers must be kept clean from dust and dew.
To verify the reliability of the $T_s$ observational records, we analyzed the
relationship between $T_a$ and $T_s$ in the observed records for 1960–2003. As shown in
Figures. S1, the mean Pearson Correlation Coefficients between $T_{s-max}$ and $T_{a-max}$
calculated from the monthly anomalies were 0.775, 0.843, and 0.806 for the annual,
warm, and cold seasonal scales, respectively, and were statistically significant (99%
confidence) for all stations. The mean correlation coefficients between $T_{s-min}$ and $T_{a-min}$
were 0.861, 0.842, and 0.865 for the annual, warm, and cold seasonal scales,
respectively, and were statistically significant (99% confidence) for all stations. The
high correlation between $T_a$ and $T_s$ indicates that the observations of $T_s$ are reliable for
detecting climate change.
SSR is the most fundamental energy resource for $T_s$ and $T_a$. Most previous studies
had used the observed SSR to analyze the relationship between the variation in SSR
and $T_a$ over Mainland China. However, sites for SSR observation were far less



numerous than those for other climatic variables, i.e., only 85 sites were used for SSR
observation in Liu et al. (2004); 90 sites were used in Li et al. (2015).

More importantly, it was found that sensitivity drifting of the instruments used for

the SSR observations led to a faster dimming rate before 1990 and that instrument
replacements from 1990 to 1993 had resulted in a falsely sharp increase in SSR (Wang,
2014; Wang et al., 2015a). The sparse distribution and low quality of SSR observations
make it difficult to quantify the variation in SSR and detect its impact on climate change.

We therefore used sunshine duration-derived SSR in this study, which is based on

an effective hybrid model developed by Yang et al. (2006). This model has subsequently
been improved (Wang et al., 2015a; Wang, 2014) and has proved to be performed well
in regional and global applications (Tang et al., 2011; Wang et al., 2012). Sunshine
duration-derived solar radiation not only can accurately reflect the impact of clouds and
aerosols on the SSR but also can more exactly reveal long-term SSR trends (Wang et
al., 2015a; Wang, 2014). Sunshine duration has a better correlation with the satellite-
derived SSR, reanalysis, and climate model simulations of SSR than the observed SSR
in China (Wang et al., 2015a).

There are 2,474 meteorological stations reporting data; however, the lengths of the

effective observation records for the stations are different. In addition, only a small





number of stations existed prior to 1960 and the observational records of $T_s$ at many
stations became significantly abnormal after 2003 because of automation. Therefore, in
our analysis, we selected 1,977 meteorological stations (see Fig. 1) that the valid data
of observation record must be longer than 30 years during the period of 43 years
between 1960 and 2003.

The monthly anomaly relative to the 1961-1990 climatology was calculated based

on a monthly mean value of the daily observation value, and if a month has more than
7 daily missing values, it was classified as a missing value (Sun et al., 2016; Li et al.,
2015). The annual anomalies are the average of the monthly anomalies for the entire
year. The anomalies in the warm seasons are the averages of the monthly anomalies
from May to October, and the anomalies in the cold seasons are the averages of the
monthly anomalies from November to the next April.
**2.2 Method**

As shown in Fig. 1, the spatial distribution of the weather stations over Mainland

China is extraordinarily asymmetric and the density of weather stations in East China
is far greater than in West China. We used the area-weight average method to reduce
these biases when calculating the national mean. First, we divided the study region into
$1° \times 1°$ grids (see Fig. S2); there are 953 grids over China. Second, we assigned all



selected stations to the grids; there are 627 grids with stations, accounting for 65.79%
of the total. Finally, the grid box value is taken to be the average of all of the stations
on the grid, and the national mean is the area-weight average of all of the effective grids
(Jones and Moberg, 2003).

The linear trends reported in this study were calculated by a linear regression based

on the least square method. Based on the anomalies of grids, there are two common
ways to calculate the national mean trends of the variables in China. The first method
(Method I) calculates the national mean monthly anomalies by taking the area-weight
of every grid first and then calculates the national mean trend based on the time series
of the national average anomalies. The second method (Method II) calculates the trend
at every grid first and then the national mean trend over China is the area-weighted
average value of the trends on all of the grids. In our study, we calculated the national
mean trends of the temperatures using both methods as both methods are widely used
in the existing studies (Gettelman and Fu, 2008). Same results are derived from those
two methods if time series of all grids is integral and have no missing data (Zhou et al.,
2009). However, as noted, we selected 1,977 stations (see Fig. 1) that the valid data of
observation records are longer than 30 years during the period 1960-2003, which is a
reasonable compromise between the integrity of the observation records and the spatial
coverage. The missing data in the time series for some grids results in a little difference





between the results of these two methods. To avoid misunderstanding, the trends
derived from Method I was discussed in the main text, but results from two methods
were shown in Table 1.

In this study, a multiple linear regression (see Eq. (1)) was used to calculate the

sensitivity and impact of changes of SSR and precipitation on the temperatures (Roy
and Haigh, 2011). This can be expressed as

$z = a \cdot x + b \cdot y + c + \varepsilon$                 (1)

where $z$ represents the monthly anomalies of $T_{s\text{-}max}$, $T_{s\text{-}min}$, $T_{a\text{-}max}$, and $T_{a\text{-}min}$; $x$ and $y$ are
the monthly anomalies of the SSR and precipitation, respectively; $a$ and $b$ are the
corresponding sensitivities of the temperatures to SSR and precipitation, respectively;
$c$ is constant term; and $\varepsilon$ indicates the residuals of the equation.

To adjust for the impact of SSR and precipitation on temperatures, we took $x$ as a

time series of SSR and $y$ as a time series of precipitation, while $a$ and $b$ are the
sensitivities of the climate variables to changes in SSR and precipitation, respectively.
The method of adjusting for the impact of SSR and precipitation is expressed as

$T_{adjusted} = T_{raw} - a \cdot x - b \cdot y$                 (2)





where $T_{adjusted}$ indicates the value of the climate variables after adjusting for the impact
of SSR and precipitation and $T_{raw}$ is the value of the climate variables in the raw data.
**3. Results**
**3.1. Trends of surface temperature and air temperature**
**3.1.1 The temporal patterns in the variabilities of the temperatures**
Figs. 2 and Figs. 3 show the long-term changes in $T_{s-max}$ and $T_{a-max}$, $T_{s-min}$ and $T_{a-min}$
$_{min}$ from 1960 to 2003 respectively. In addition to annual variability (Figs. 2a and Figs.
3a), we analyzed the variabilities of the temperatures in both the warm seasons (May-
October) (Figs. 2b and Figs. 3b) and the cold seasons (November to the following April)
(Figs. 2c and Figs. 3c). In the annual records, all of the temperatures showed an obvious
warming trend over China (Figs. 2a and Figs. 3a). As shown in Table 1, the national
mean warming rate for $T_{s-max}$ was 0.227 °C $10yr^{-1}$ and the rate for $T_{a-max}$ was 0.167 °C
$10yr^{-1}$ from 1960 to 2003. The warming rate of $T_{a-max}$ based on the 1,977 stations in
this paper was a little higher than both that of the global average (0.141 °C $10yr^{-1}$) from
1950 to 2004 (Vose et al., 2005) and that of a previous analysis of China (0.127 °C
$10yr^{-1}$) from 1955 to 2000 based on 305 stations (Liu et al., 2004).
The seasonal contrasts of warming of $T_{a-max}$ and $T_{s-max}$ are important. $T_{s-max}$ had an





average rate of 0.172 °C $10yr^{-1}$ in the warm seasons and 0.354 °C $10yr^{-1}$ in the cold
seasons. For $T_{a-max}$, it was 0.091 °C $10yr^{-1}$ and 0.294 °C $10yr^{-1}$ in the warm and cold
seasons, respectively. The increases in $T_{s-max}$ and $T_{a-max}$ in the cold seasons were much
larger than those in the warm seasons, which is consistent with previous studies of
China and other regions (Shen et al., 2014; Vose et al., 2005; Ren et al., 2005).

Similarly, the warming rates of $T_{s-min}$ and $T_{a-min}$ in the warm seasons were clearly

lower than those in the cold seasons too. As shown in Figs. 3, $T_{s-min}$ increased by
0.315 °C $10yr^{-1}$ and $T_{a-min}$ increased by 0.356 °C $10yr^{-1}$ (see Figs. 3a) from 1960 to
2003. The warming trend of $T_{a-min}$ is generally consistent with earlier studies (Shen et
al., 2014; Li et al., 2015; Liu et al., 2004); however, it is considerably larger than that
reported for the global average (0.204 °C $10yr^{-1}$) (Vose et al., 2005). $T_{s-min}$ increased at
a rate of 0.221 °C $10yr^{-1}$ in the warm seasons and 0.447 °C $10yr^{-1}$ in the cold seasons
from 1960 to 2003. $T_{a-min}$ increased at rates of 0.245 °C $10yr^{-1}$ and 0.505 °C $10yr^{-1}$ in
the warm and cold seasons, respectively.

On a national average scale, all temperatures increased from 1960 to 2003. The

warming rate of $T_{s-min}$ ($T_{a-min}$) was significantly faster than that of $T_{s-max}$ ($T_{a-max}$) and the
warming rates of all temperatures in cold seasons were generally higher than those in
warm seasons. These basic characteristics of the temperature changes are consistent





with previous studies on global or regional scales (Li et al., 2015; Liu et al., 2004;
Easterling et al., 1997). For the daily contrast in warming rates, there are still apparent
uncertainties in its causes and physical mechanism (Hartmann et al., 2013). Some
previous studies hold an opinion that the microclimate (e.g. urban heat island) has a
larger impact on minimum temperatures due to the lower and more stable boundary
layer at night (Zhou and Ren, 2011; Christy et al., 2009), while many investigators
argued that the variabilities of surface solar radiation are the main reason for this daily
contrast in warming rates (Sanchez-Lorenzo and Wild, 2012; Makowski et al., 2009).
For seasonal contrast in warming rates of temperatures, previous studies attributed that
the most precipitation are concentrated in warm season because of monsoonal climate
in China. The higher relative humidity in warm seasons will suppress changes of
temperatures (Liu et al., 2004; Karl et al., 1993). In this paper, our results had partly
explained the causes of those daily or seasonal contrast in warming rates of
temperatures.

However, there remain slight differences between our results and previous studies

with respect to the temperature warming rates, which might have several causes. The
number of stations used in our study is much greater than in previous studies, which
has led to better spatial coverage and a better representation of our analytical result. In
addition, we used the area-weight average method both to reduce the impact of uneven



station densities and to improve the representation of West China, which has a sparse
station distribution (see Fig. 1), when calculating the national mean trend. Previous
studies had indicated that the Tibet Plateau and the northwestern arid and semiarid
regions were experiencing significantly more rapid warming trends than other regions
in China (You et al., 2016), which may lead to our results being slightly higher than
those of previous studies. In addition, the study periods of the various studies are not
exactly the same. In addition, differences in the method used to calculate the national
mean trend also may result in significant differences. As shown in Table 1, the absolute
value of difference between Method I and Method II ranges from 0.011 to 0.033 °C
$10yr^{-1}$, account for 3.4% to 14.3% of trends (taking the results of Method I as reference).
**3.1.2. The spatial patterns in the variabilities for the temperatures**

Figs. 4 demonstrates a clear spatial heterogeneity in the warming rates for $T_{s-max}$

and $T_{a-max}$ over China from 1960-2003. $T_{s-max}$ and $T_{a-max}$ increased at high rate and the
trends of $T_{s-max}$ and $T_{a-max}$ were statistically significant in the Tibet Plateau, and
Northwest and Northeast China (see Figs. S3). However, $T_{s-max}$ and $T_{a-max}$ had a relative
lower warming rate in the North China Plain and South China, and $T_{s-max}$ even showed
cooling trends in the Sichuan Plain, the Yangtze River Delta, and the Pearl River Delta.
Lower warming rates of $T_{a-max}$ in South China and the North China Plain had also



been reported in multiple previous studies (Liu et al., 2004; Li et al., 2015).
For $T_{s-max}$ and $T_{a-max}$, the warming rates of South China and the North China Plain
in the warm seasons were considerably lower than those in the cold seasons, resulting
in a more obvious spatial heterogeneity in the warm seasons (Figs. 4b and 4h). However,
the warming rates of both $T_{s-max}$ and $T_{a-max}$ in the Sichuan Basin and the Pearl River
Delta were lower in the cold seasons than in the warm seasons. Despite of the spatial
and seasonal patterns of $T_{a-max}$ were clearly similar to those of $T_{s-max}$. For $T_{a-max}$, both
the seasonal asymmetry and the spatial heterogeneity of the warming trend were less
than those of $T_{s-max}$.
For $T_{s-min}$ and $T_{a-min}$, the warming rates were highest in North China and generally
decreased from north to south (Figs. 4d and 4j). The average warming rates of $T_{s-min}$
and $T_{a-min}$ in the cold seasons (Figs. 4f and 4l) were faster than those in the warm seasons
(Figs. 4e and 4k). This variation of warming rate with latitudes have been attributed to
dynamics amplification (Wallace et al., 2012; Ding et al., 2014). In this study, we focus
on the spatial heterogeneity of the warming rates at similar latitudes and diurnal contrast
of the warming rates.
By contrasting the annual variation and spatial pattern of trends, we found that $T_s$
and $T_a$ had an extremely significant correlation with each other. Based on the time series



of national mean yearly anomalies (see Figs. 2 and Figs. 3), the correlations between
$T_{s-max}$ and $T_{a-max}$ were 0.877, 0.799, and 0.921 on the annual, warm, and cold seasonal
scales, respectively. The correlations between $T_{s-min}$ and $T_{a-min}$ were 0.976, 0.969, and
0.977 on the annual, warm, and cold seasonal scales, respectively. In the spatial pattern
of the trends (Figs. 4), the correlations between $T_{s-max}$ and $T_{a-max}$ were 0.488, 0.465, and
0.522 on the annual, warm, and cold seasonal scales, respectively. Those between $T_{s-min}$
$_{min}$ and $T_{a-min}$ were 0.638, 0.670, and 0.594 on the annual, warm, and cold seasonal
scales, respectively.

In summary, $T_s$ had a significant correlation with Ta both in annual variation (Figs.

2 and Figs. 3) and in long-term trends (Figs. 4), indicating that $T_s$ observational records
are reliable for climate change research. However, the correlation between $T_{s-min}$ and
$T_{a-min}$ was significantly higher than that between $T_{s-max}$ and $T_{a-max}$. $T_{s-min}$ is closely
related to the land–atmosphere longwave wave radiation balance during the nighttime,
which is closely related to the atmospheric greenhouse effect (Dai et al., 1999). During
the daytime, $T_s$ is directly determined by the land surface energy balance, i.e., the
incoming energy including SSR and atmospheric longwave radiation (Wang and
Dickinson, 2013a), and its partitions into latent and sensible heat fluxes (Zhou and
Wang, 2016). Despite its dependence on the land-atmosphere sensible heat flux, $T_a$ is
also impacted by local and/or large-scale circulation. So, the changes of land surface





energy balance caused by SSR and precipitation have different levels of effect on $T_s$
and $T_a$ during the day, which most likely causes a lower correlation between $T_{s-max}$ and
$T_{a-max}$ than that between $T_{s-min}$ and $T_{a-min}$.
**3.2. The impact of surface solar radiation and precipitation on temperatures**
**3.2.1 Impact of surface solar radiation**

Figs. S4 shows that SSR had an important relationship with $T_{s-max}$ and $T_{a-max}$ but

not with $T_{s-min}$ and $T_{a-min}$. The national mean of the partial correlation coefficients
between SSR and $T_{s-max}$ is 0.552 and 98.9% of the stations are statistically significant
at the 1% level. Meanwhile, the national mean of the partial correlation coefficients
between SSR and $T_{a-max}$ is 0.441, and 95.4% of the stations are statistically significant
at the 1% level. This relationship is stronger in South China and on the North China
Plain, i.e., it reaches 0.810 for $T_{s-max}$ and 0.765 for $T_{a-max}$.

$T_{s-max}$ is primarily determined by the land surface energy budget, whereas $T_{a-max}$ is

influenced by the land surface energy flux and other factors, including both large-scale
circulation and anthropogenic heat flux. Therefore, the correlation between $T_{s-max}$ and
SSR is higher than that between $T_{a-max}$ and SSR (Figs. S4a and   Figs. S4g). On the
seasonal scales, the partial correlation between $T_{s-max}$ ($T_{a-max}$) and SSR in warm seasons





is higher than that in cold seasons, and the national mean partial correlation coefficients
for the warm and cold seasons are 0.579 and 0.498 for $T_{s-max}$ and 0.544 and 0.386 for
$T_{a-max}$, respectively, consisting with the seasonal cycle of SSR intensity over China.

Spatially, overall, the partial correlation coefficients between $T_{s-max}$ and $T_{a-max}$ and

SSR are higher in South China than in North China (see Figs. S4a–c and   Figs. S4g–i).
South of 35° N, the national mean of the partial correlation coefficients between $T_{s-max}$
($T_{a-max}$) and SSR is 0.654 (0.552), whereas that between $T_{s-max}$ ($T_{a-max}$) and SSR is just
0.417 in north of 35° N. During daytime, $T_s$ and $T_a$ is largely determined by how much
energy is used to evapotranspiration (Shen et al., 2014). In south China where soil
moisture is high, the energy used for evapotranspiration is near linearly related to SSR
(Wang and Dickinson, 2013b; Zhou et al., 2007). However, energy used for
evapotranspiration is more dependent on precipitation in the northwest China where the
soil is dry during most time of a year. As a result, the energy available for heating
surface and air temperature is not so closely related SSR. Therefore, the correlation
coefficients between SST and $T_{s-max}$ (or $T_{a-max}$) were stronger in south China. The
correlation between $T_{a-max}$ and SSR along the coast was significantly lower than inland
(see Figs. S4g-i), especially in the cold seasons (see Figs. S4i), probably because of the
stronger land-sea thermal convection along the coast.





To quantify the impact of SSR on temperature, the sensitivity of temperatures to
changes in SSR has been calculated (Eq. (2)). As Figs. S5 shows, $T_{s\text{-}max}$ was the most
sensitive to SSR, followed by $T_{a\text{-}max}$, and their national means were 0.092 °C $(W\,m^{-2})^{-1}$
and 0.035 °C $(W\,m^{-2})^{-1}$, respectively. $T_{s\text{-}min}$ and $T_{a\text{-}min}$ were insignificantly sensitive to
SSR because they primarily depend on atmospheric longwave radiation during the
nighttime.
Based on the above analysis, we calculated the impact of changes in SSR on
temperature (see the Method Section). From 1960–2003, the national mean decreasing
rate of SSR was −1.502 W m$^{-2}$ 10yr$^{-1}$, as calculated from monthly anomalies at 1,977
stations, and the trend was significant in most regions over China (see Figs. S6).   Our
results are considerably less than the global average dimming rate (−2.3 ~ −5.1 W m$^{-2}$
10yr$^{-1}$) between the 1960s and the 1990s (Gilgen et al., 1998; Liepert, 2002; Stanhill
and Cohen, 2001; Ohmura, 2006) and the national mean dimming rate across China
(−2.9 ~ −5.2 W m$^{-2}$ 10yr$^{-1}$) between the 1960s and the 2000s based on radiation station
observations (Che et al., 2005; Liang and Xia, 2005; Shi et al., 2008; Wang et al., 2015a).
As noted in data section, the sensitivity drifting and replacement of the instruments
used for the SSR observations results in a significant homogenization in stations
observation records (Wang, 2014; Wang et al., 2015a), which causes a great uncertainty





in trend estimation. Tang et al. (2011) used quality-controlled observational data from
72 stations and two radiation models based on 479 stations to determine both that the
dimming rate over China is $-2.1 \sim -2.3$ W m$^{-2}$ 10yr$^{-1}$ during 1961-2000 and that the
SSR has been essentially unchanged since 2000; this finding is generally consistent
with our results.
Due to the decreasing trend in SSR, the warming trend of $T_{s\text{-}max}$ and $T_{a\text{-}max}$
decreased by 0.139 °C 10yr$^{-1}$ and 0.053 °C 10yr$^{-1}$, respectively, in the national mean.
Spatially, the decreasing rate of SSR in South China and the North China Plain was
significantly higher than in other regions, especially in the warm seasons (Figs. 5b).
Therefore, the cooling effect of decreasing SSR on $T_{s\text{-}max}$ and $T_{a\text{-}max}$ was more
significant in South China and the China North Plain, resulting in significantly lower
warming rates of $T_{s\text{-}max}$ and $T_{a\text{-}max}$ there than in other regions (see Figs. 4). The spatial
consistency between decreasing SSR and the warming slowdown of $T_{s\text{-}max}$ ($T_{a\text{-}max}$)
implies that variation in SSR is the primary reason for the spatial heterogeneity of the
warming rate in $T_{s\text{-}max}$ ($T_{a\text{-}max}$).
**3.2.2 Impact of Precipitation**
Figs. S7a shows that there is a significant negative correlation between $T_{s\text{-}max}$ and
precipitation; the national mean of the partial correlation coefficients is $-0.323$, and





99.3% of the stations are statistically significant at the 1% level. Seasonally, the
correlation is stronger in the warm seasons (regional mean: −0.405) than in the cold
seasons (regional mean: −0.276). In warm seasons, the correlation in North China
(regional mean: −0.459) is clearly stronger than in South China (regional mean: −0.365).
In cold seasons, the correlation is highest on the Southwestern Yunnan-Guizhou Plateau
and in most regions of North China (regional mean: −0.305) (Figs. S7b and Figs. S7c),
whereas it is relatively weak in Southeastern China, the Tibet Plateau, Dzungaria, the
Tarim Basin, and some regions of Northeastern China (regional mean: −0.117). The
correlations between $T_{a-max}$ and precipitation have similar spatial and seasonal patterns
(Figs. S7g–i) too, and 35.4% of the stations are statistically significant at the 1% level;
these are primarily concentrated in arid and semiarid regions of China (regional mean:
−0.167) (Figs. S7e–f and Figs. S7j–l).
Precipitation has a negative relationship with temperature because precipitation
can reduce temperatures by increasing surface evaporative cooling (Dai et al., 1997;
Wang et al., 2006). The impact of precipitation on temperature is higher in the warm
seasons over China, which is consistent with seasonal changes in the correlation
between $T_{s-max}$ and $T_{a-max}$ and precipitation (see Figs. S7b–c and Figs. S7h–i).
The national mean sensitivities of $T_{s-max}$ and $T_{a-max}$ to precipitation were −0.321 °C



10 mm$^{-1}$ and $-0.064$ °C 10 mm$^{-1}$, respectively. As shown in Figs. S8, there were
apparent seasonal and spatial changes in the sensitivity of $T_{s\text{-}max}$ and $T_{a\text{-}max}$ to
precipitation (Figs. S8a–c and Figs. S8g–i). In warm seasons, these sensitivities were
highest in the Tibet Plateau, the Loess Plateau, the Inter Mongolia Plateau, Dzungaria,
and the Tarim Basin (Figs. S8b and Figs. S8h). In cold seasons, the distribution of
regions with high sensitivity extended to all of North China and Southwest China (Figs.
S8c and Figs. S8i). Overall, the sensitivities of $T_{s\text{-}max}$ ($T_{a\text{-}max}$) were significantly higher
in arid regions (dry seasons) than in humidity regions (rainy seasons) (Wang and
Dickinson, 2013b). In contrast, $T_{s\text{-}min}$ and $T_{a\text{-}min}$ were less sensitive to variations in the
precipitation.

As Figs. 6 shows, during 1960-2003, the trend in the precipitation over the 1977

stations had obvious spatial heterogeneities. China's precipitation during this period
showed a slight increasing trend with an increasing rate of 0.112 mm 10yr$^{-1}$.
Precipitation in Northwestern China and Southeastern China experienced an increasing
trend, whereas precipitation in the North China Plain, the Sichuan Basin, and parts of
Northeastern China experienced a decreasing trend. However, the trend of precipitation
was insignificant in most regions (see Figs. S6). Variation in precipitation had
significant seasonal differences (see Figs. 6b and Figs. 6c). The seasonal and spatial
characteristics of these precipitation variations are consistent with those identified in





previous studies (Zhai et al., 2005; Wang et al., 2015b).

Therefore, for $T_{a\text{-max}}$ and $T_{s\text{-max}}$, the reduction in precipitation aggravated the

warming trend in the North China Plain, Sichuan Basin, and parts of Northeastern China,
whereas the increase in precipitation primarily slowed the warming trend in
Northwestern China and on the Mongolian Plateau (Figs. 6d). On national average, the
impact of increasing precipitation resulted in the warming trends of $T_{s\text{-max}}$ and $T_{a\text{-max}}$
being decreased by $-0.007$ °C $10yr^{-1}$ and $-0.002$ °C $10yr^{-1}$, respectively. However,
compared to SSR, the impact of precipitation on $T_{s\text{-max}}$ was smaller by approximately
an order of magnitude. For $T_{s\text{-min}}$ and $T_{a\text{-min}}$, the impact of changes in precipitation was
insignificant.
**3.3. Trends of surface and air temperature after adjusting for the effect of SSR**
**and precipitation**

Based on the above analysis of the impact of SSR and precipitation on

temperatures, we found that the variation of SSR and precipitation had little effect on
$T_{s\text{-min}}$ and $T_{a\text{-min}}$. Therefore, we only analyzed their impact on $T_{s\text{-max}}$ and $T_{a\text{-max}}$. After
adjusting for the impact of SSR and precipitation (Figs. 7), the warming rates of $T_{s\text{-max}}$
and $T_{a\text{-max}}$ increased by 0.146 °C $10yr^{-1}$ (64.3%) and 0.055 °C $10yr^{-1}$ (33.0%),
respectively.





After adjusting, the seasonal contrast warming rates of $T_{s\text{-}max}$ and $T_{a\text{-}max}$ decreased
by 45.0% and 17.2%. The national mean warming rate of $T_{s\text{-}max}$ increased by 0.178 °C
$10yr^{-1}$ (103.1%) in the warm seasons and 0.086 °C $10yr^{-1}$ (27.2%) in the cold seasons.
For $T_{a\text{-}max}$, the warming rate increased by 0.069 °C $10yr^{-1}$ (76.4%) in the warm seasons
and 0.034 °C $10yr^{-1}$ (11.7%) in the cold seasons.
After adjusting for the impact of SSR and precipitation, the difference in warming
rates between $T_{a\text{-}max}$ and $T_{a\text{-}min}$ changed from 0.190 to 0.134 °C $10yr^{-1}$, a decrease of
29.1%, and the difference between $T_{s\text{-}max}$ and $T_{s\text{-}min}$ changed from 0.088 to 0.058 °C
$10yr^{-1}$, a decrease of 33.0%.
More importantly, after adjusting for the impact of SSR and precipitation, the
spatial coherence of the warming rates of $T_{s\text{-}max}$ and $T_{a\text{-}max}$ in South and North China
clearly improved (Figs. 8). The regional difference between the North China Plain,
South China, and other regions in China shrank significantly due to the increase in the
warming rates in South China and the North China Plain. In addition, the warming trend
of $T_{s\text{-}max}$ and $T_{a\text{-}max}$ became more statistical significant in North China Plain and South
China (see Figs. S9).
To further prove this, we selected two regions in China for further investigation:
R1 primarily includes the North China Plain and R2 primarily includes the Loess





Plateau, as shown in Figs. 9a. These regions share same latitudes. However, the SSR
showed substantially contrasting trends in the two regions (see Figs. 9b). After
adjusting for the impacts of SSR and precipitation, the annual trends of $T_{s\text{-max}}$ and $T_{a\text{-max}}$
in R1 increased by 0.304 and 0.118 °C $10\text{yr}^{-1}$, while those in R2 just increased by
0.025 and 0.016 °C $10\text{yr}^{-1}$. The difference in warming rates of $T_{s\text{-max}}$ and $T_{a\text{-max}}$ between
R1 and R2 reduced significantly after adjusting (see Figs. 9d).

Meanwhile, in R1, the seasonal and diurnal difference in the warming rates of $T_{s\text{-max}}$

and $T_{a\text{-max}}$ decreased significantly. After adjusting, in R1, the difference in warming
rates between warm seasons and cold seasons decreased by 68.7% for $T_{s\text{-max}}$ and
decreased by 50.8% for $T_{a\text{-max}}$. The difference in warming rates between $T_{s\text{-max}}$ and $T_{s\text{-min}}$
decreased by 93.4% and that between $T_{a\text{-max}}$ and $T_{a\text{-min}}$ decreased by 59.6%. The
seasonal and diurnal difference of temperatures in R2 had no significant changes after
adjusting. All in all, the trends of R1 and R2 became more consistent with each other
after adjusting the difference in SSR and precipitation between them (see Figs. 9d).
**4. Conclusions and Discussion**

In China, despite the general warming trends over the entire country, warming

trends showed significant spatial and temporal heterogeneity. In this paper, we analyzed
the spatial and temporal patterns of Ts and Ta from 1960 to 2003 and further analyzed





and quantified the impact of SSR and precipitation on the temperature. The main results
are as follows.

The national mean warming rates of $T_{s\text{-max}}$, $T_{s\text{-min}}$, $T_{a\text{-max}}$, and $T_{a\text{-min}}$ were 0.227,

0.315 °C $10\text{yr}^{-1}$, 0.167 °C $10\text{yr}^{-1}$, and 0.356 °C $10\text{yr}^{-1}$, respectively, from 1960 to 2003.
The warming rates of $T_{s\text{-min}}$ and $T_{a\text{-min}}$ were significantly greater than those of $T_{s\text{-max}}$ and
$T_{a\text{-max}}$ (see Figs. 2 and Figs. 3). Warming rates of $T_{s\text{-max}}$ and $T_{a\text{-max}}$ in South China and
on the North China Plain were significantly lower than other regions (see Figs. 4). The
spatial heterogeneity in the warm seasons was greater than in the cold seasons.

During the study period, SSR decreased by $-1.502$ W m$^{-2}$ $10\text{yr}^{-1}$ in China, with

higher dimming rates in South China and the North China Plain. Using partial
regression analysis, we found that SSR was the primary cause of the spatial pattern in
the warming rates of $T_{s\text{-max}}$ and $T_{a\text{-max}}$. After adjusting for the impact of SSR and
precipitation, the warming trend of $T_{s\text{-max}}$ increased by 0.146 °C $10\text{yr}^{-1}$ and that of $T_{a\text{-}}$
$_{\text{max}}$ increased by 0.055 °C $10\text{yr}^{-1}$. After adjustments, the trends of $T_{s\text{-max}}$, $T_{s\text{-min}}$, $T_{a\text{-max}}$,
and $T_{a\text{-min}}$ became 0.373 °C $10\text{yr}^{-1}$, 0.315 °C $10\text{yr}^{-1}$, 0.222 °C $10\text{yr}^{-1}$, and 0.356 °C
$10\text{yr}^{-1}$. The reduction of SSR resulted in the warming rates of $T_{s\text{-max}}$ and $T_{a\text{-max}}$
decreasing by 0.139 °C $10\text{yr}^{-1}$ and 0.053 °C $10\text{yr}^{-1}$, accounting for 95.0% and 95.8%,
respectively, of the total impact of SSR and precipitation.




In addition to SSR and precipitation, temperatures' warming rates may be affected

by many other factors, such as land cover change, that have not been discussed in this

study due to lack of data, i.e., land cover and land use (Liu et al., 2005; Zhang et al.,

2016). After adjusting for the impact of SSR and precipitation changes, spatial

differences in the warming trends clearly decreased; however, some regional

differences remain. The warming rate of $T_{s-max}$ in the Sichuan Basin remained

significantly lower than in other regions after adjusting for these impacts. In addition,

the north-south difference in the warming rates of $T_{s-min}$ and $T_{a-min}$ cannot be explained

by the impacts of SSR and precipitation. Further study is needed.

**Acknowledgements** This study was funded by the National Natural Science

Foundation of China (41525018 and 91337111) and the National Basic Research

Program of China (2012CB955302). The latest meteorological datasets, collected at

approximately 2400 meteorological stations in China were obtained from the China

Meteorological Administration (CMA, http://data.cma.gov.cn/data).



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





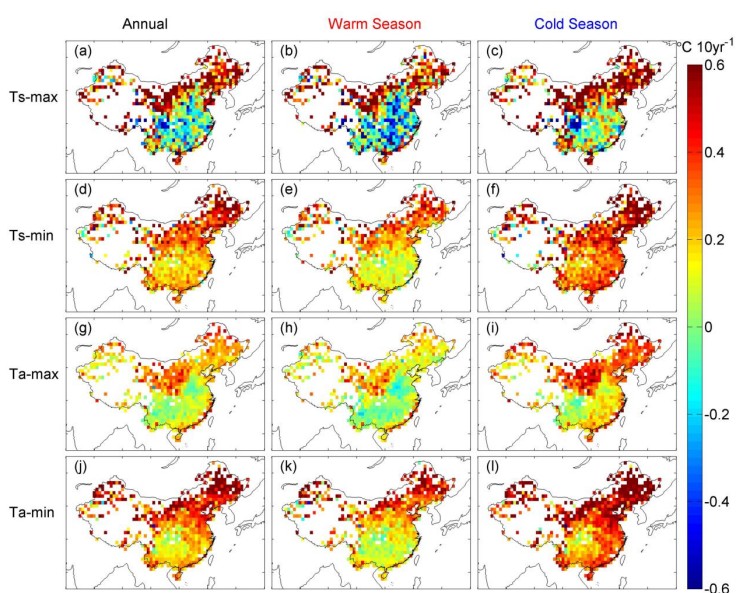


Figs. 4. Maps of the trends of the monthly anomalies for $T_{s\text{-}max}$ (a–c), $T_{s\text{-}min}$ (d–f), $T_{a\text{-}max}$
(g–i), and $T_{a\text{-}min}$ (j–l) on annual, warm (May–October), and cold (November–next April)
seasonal scales.





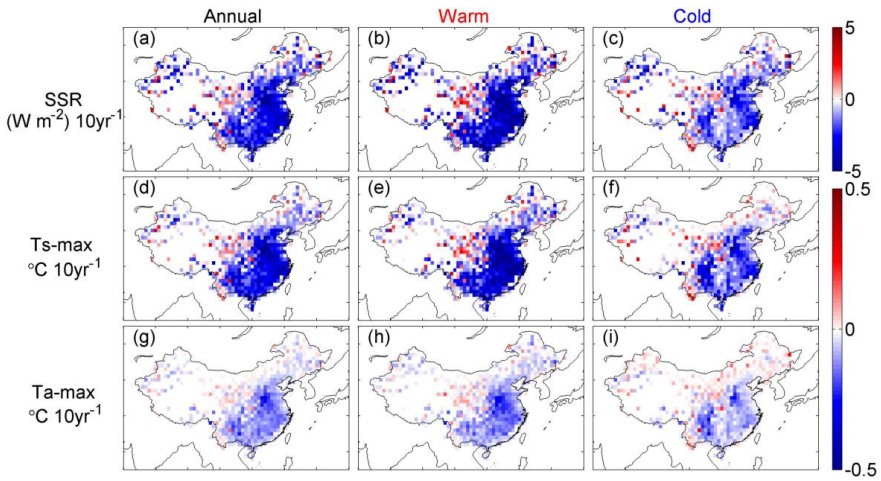


Figs. 5. Maps of the trends in surface solar radiation (SSR) (a–c) and its impact on the
warming rates of $T_{s\text{-}max}$ (d–f) and $T_{a\text{-}max}$ (g–i).




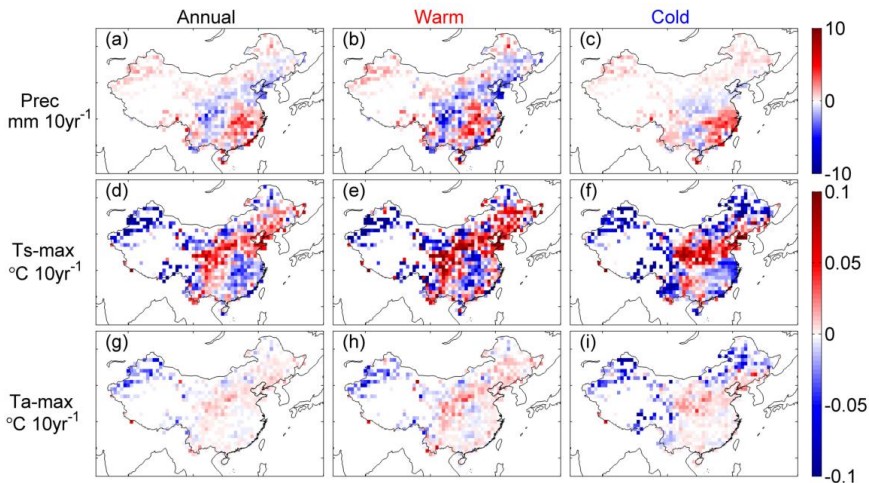


Figs. 6. Maps of the trends in precipitation (Prec) (a–c) and their impact on the warming
rates for $T_{s-max}$ (d–f) and $T_{a-max}$ (g–i).




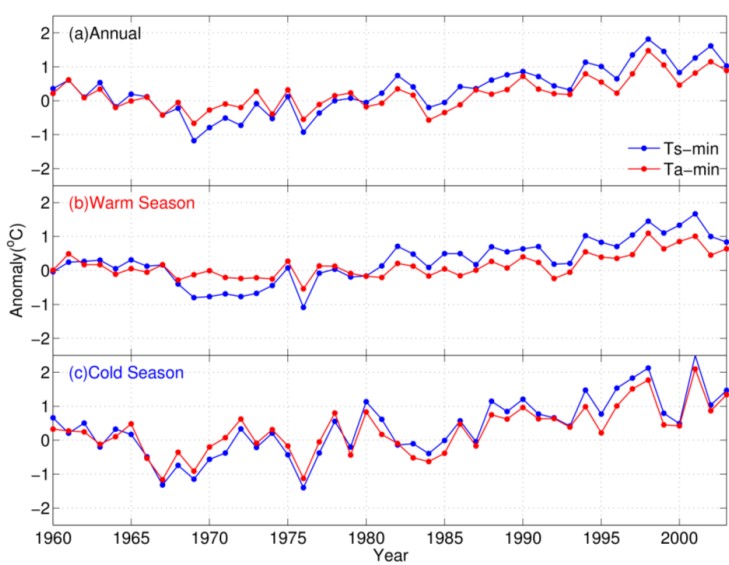


Figs. 7. After adjusting for the impact of SSR and precipitation, regionally average
anomalies of $T_{s-max}$ (blue line) and $T_{a-max}$ (red line) on annual (a), warm (b), and cold (c)
seasonal scales for the reference period of 1961-1990.





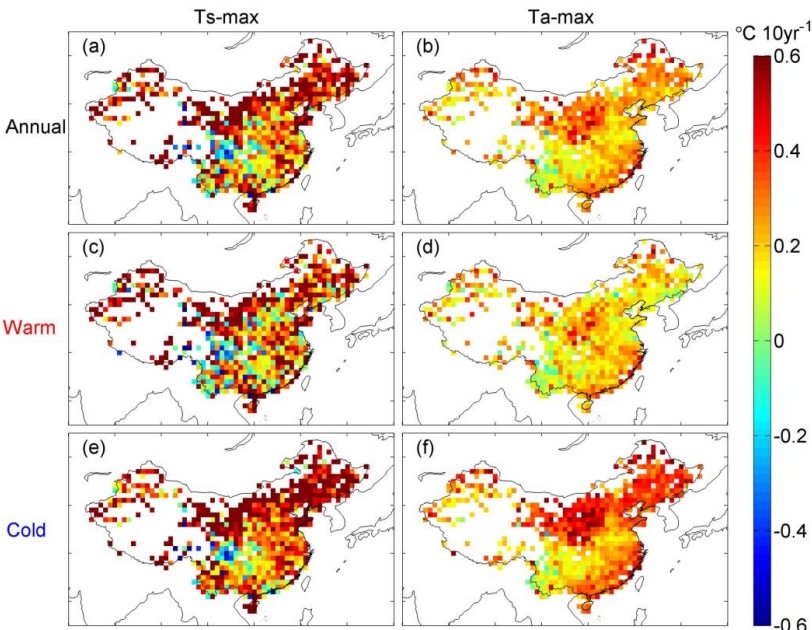


Figs. 8. After correcting for the impact of solar radiation and precipitation, maps of the
trends of the monthly anomalies for $T_{s-max}$ (a, c, e) and $T_{a-max}$ (b, d, f) on the annual,
warm, and cold seasonal scales.





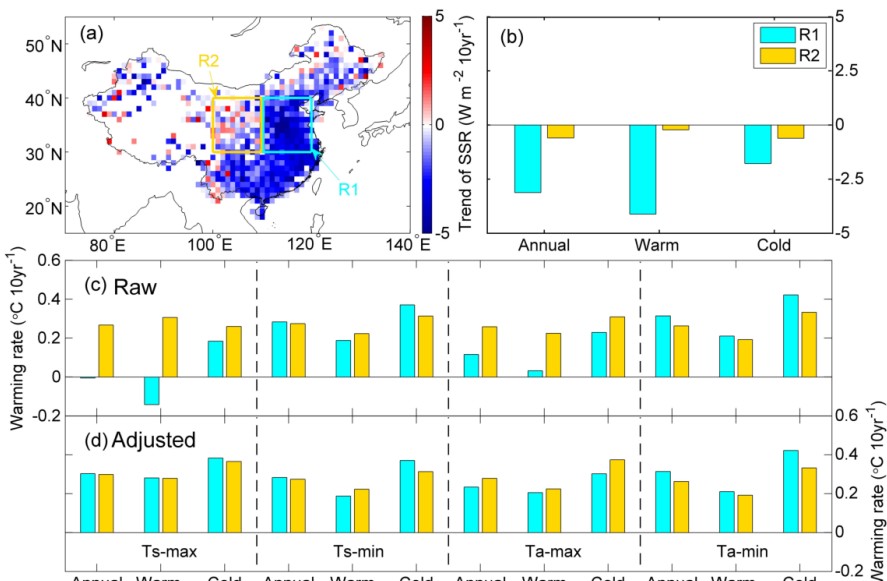


Figs. 9. (a) Maps of the trends of the surface solar radiation (SSR) and the location of
the selected regions, R1 (latitude: 30° N–40° N; longitude: 110° N–120° N) and R2
(latitude: 30° N–40° N; longitude: 100° N–110° N). (b) The national mean trends of R1
and R2. (c) The trends on the annual, warm, and cold seasonal scales calculated based
on the raw data (Raw). (d) The trends on the annual, warm, and cold seasonal scales
calculated based on the adjusted data (Wang et al., 2015a), which does not include the
impact of the surface solar radiation variation.




Table 1. The warming rates (units: °C 10yr$^{-1}$) of the temperatures on annual, warm, and
cold seasonal scales. Raw and Adjusted represent the warming rates calculated for the
data before and after adjusting for the impact of solar radiation and precipitation.
Method I represents the first method, which calculates the national mean anomalies first
and then calculates the national mean trend based on this time series; Method II
represents second method, which calculates the trend of every grids first and then
calculates the national mean value of the trends of all grids using the area-weight
average method. We calculated the national mean trends of the temperatures using both
methods.

|  |  |  | $T_{s-max}$ | $T_{s-min}$ | $T_{a-max}$ | $T_{a-min}$ |
|---|---|---|---|---|---|---|
| Method I | Raw | Annual | 0.227 | 0.315 | 0.167 | 0.356 |
|  |  | Warm | 0.172 | 0.221 | 0.091 | 0.245 |
|  |  | Cold | 0.354 | 0.447 | 0.294 | 0.505 |
|  | Adjusted | Annual | 0.373 | - | 0.222 | - |
|  |  | Warm | 0.350 | - | 0.160 | - |
|  |  | Cold | 0.450 | - | 0.329 | - |
| Method II | Raw | Annual | 0.254 | 0.328 | 0.183 | 0.368 |
|  |  | Warm | 0.193 | 0.235 | 0.104 | 0.256 |
|  |  | Cold | 0.321 | 0.415 | 0.264 | 0.476 |
|  | Adjusted | Annual | 0.401 | - | 0.239 | - |
|  |  | Warm | 0.374 | - | 0.174 | - |
|  |  | Cold | 0.432 | - | 0.304 | - |
