# Peer review of "Atmos. Chem. Phys. Discuss., doi:10.5194/acp-2016-1022, 2016 Manuscript under review for journal Atmos. Chem. Phys."

_Atmospheric Chemistry and Physics, 2016_

## Referee Comment (RC1) · A. K. Betts (Referee) · 2 Jan 2017

General Comments

This is an important paper with interesting Figures on the regional variation of surface temperature trends over China, and their relation to regional precipitation and SSR.

The biggest challenge for this reviewer is that I unsure exactly what the elegant Figures show. There are critical gaps between the methods section and the Figures. The legends rather than the text try to explain the content of the Figures, and they are written for the authors, not for a global audience, which will struggle to follow the missing steps

in the logic. The reader cannot connect the symbols in Methods to the symbols in the Figures, and the description of the Figures.

Technical details

Methods uses Traw and Tadjusted and monthly anomalies, as well as 'Z' for a regression fit to monthly anomalies of T. Do all the graphs show anomalies? Which ones show Tadjusted? Which ones show regression fits Z?

Eq (1), 2 and 5 are just textbook definitions, which are poorly defined for this specific analysis. They use 'a' and 'b' as symbols for different coefficients in 1, 2 and 5. The values for these (a, b) in this analysis may appear in later figures, but the reader has to guess how they were actually computed. Which Figs show which coefficients or adjusted variables is unclear, because they are largely labeled the same: eg Ts-max or Ta-min, or just 'PC'.

Relabel PCa, PCb, PCc, PCd etc with a clear connection to a numbered equation coefficient. Use the same specific language to describe the coefficient in both methods and text introducing the Figure.

Consider adding a simple label to distinguish Tadjusted from T in the Figures.

L177-180 Comment that the number of sunshine duration stations (105 in Wang et al. 2015a) is still small compared with the Ta data. How well are they distributed in western China?

L242 What are the coefficients a and b; and their uncertainties? Cross-reference where you show these. When you reach Figs 5 and 6, it is unclear how they relate to Eq (2)

L245 There are no equations 3 and 4.

L251 And Figs 2 and 3. Are these Traw or Tadjusted?

Section 3.1.1 and Table 1 All these results are presented as mean trends with no estimate of uncertainty. Add some error estimates.

Section 3.2.1 You need an explicit explanation of Fig 5 and then 6, The reader cannot see clearly how they were constructed. What are these partial correlation coefficients using precipitation as control? Do they relate to the Tadjusted in (5) or the sensitivities in (2). Nothing has been defined or connected logically (and Eq (3) and (4) are missing? Same issues for Figure 8 and 9.

Fig 6 Is this the coefficient 'a' in Eq(2)? Where do you show coefficient 'b'? Is it in Fig 9?

Fig 11 Is this the first time Tadjusted is plotted?

L136 and L770 cite different references for the dataset

Language issues

The structuring of sentences is generally very good, but verbs and tenses need occasional editing, but I will leave this to later editing. An example is 106 LST... plays an important role in climate change 107 research because it directly relates to the land surface energy budget. Previously, Ts 108 values used in regional climate research were primarily derived

---

## Referee Comment (RC2) · Anonymous Referee #2 · 9 Jan 2017

This paper analysed the spatial patterns of Ts and Ta and their relations with SSR and precipitation using the observations. It is important to study the mechanism of T changes in the warming climate in regional scales. I think this article is publishable after major corrections.

In general, the text can be compressed substantially.

1) Eq (1) is not needed, "linear trend" or "Linear regression" should be enough.

2) Please discuss why the Tadjusted is calculated? State its actual meaning and applications.

3) The detailed descriptions are not necessary in Figure captions and can be moved to the text.

Minor

Line 54: Hegerl and Zwiers is missing in the references

Line 173-174: Is that 1990?

Line 257: Please check Ts-max and Ta-max trends. By eye, both values should be close.

Line 277-281: Mechanism of the difference should be mentioned here.

Line 294: Significant difference. Can you clarify how significant it is please?

Line 534: References

Line 570: Eastling et al and Line 638: Ohmura. They are not referenced in the text, please check.

Ta-min in Figs 5,6,8,9 can be removed, since they don't give much information. They can be briefly discussed in the text.

---

## Author Comment (AC1) · 20 Jan 2017

Please find our response below your comments

1) General Comments This is an important paper with interesting Figures on the regional variation of surface temperature trends over China, and their relation to regional precipitation and SSR. The biggest challenge for this reviewer is that I unsure exactly what the elegant Figures show. There are critical gaps between the methods section and the Figures. The legends rather than the text try to explain the content of the Figures, and they are written for the authors, not for a global audience, which will struggle

to follow the missing steps in the logic. The reader cannot connect the symbols in Methods to the symbols in the Figures, and the description of the Figures.

Reply: Thank you for the high recommendation and constructive comments. We have carefully checked and revised logical structure of paper and unified the symbols for Methods and Figures. Below please find our point to point response to your comments.

2) Technical details Comments: Methods uses Traw and Tadjusted and monthly anomalies, as well as 'z' for a regression fit to monthly anomalies of T. Do all the graphs show anomalies? Which ones show Tadjusted? Which ones show regression fits 'z'?

Reply: In this study, all of trends and regression analyses are based on the monthly anomalies of temperatures (T, including Ts-max, Ts-min, Ta-max, Ta-min), surface solar radiation (Rs) and precipitation (P) during 1960-2003. We explicitly claimed in Lines 213-214: "The linear trends reported in this study were calculated by a linear regression based on monthly anomalies of T, Rs, and P" and in Lines 229-230: "The impact of Rs/P on Ts-max/Ta-max is calculated by a multiple linear regression (Roy and Haigh, 2011) from monthly anomalies.". In this revised paper, we deleted the Eqs. (1) and (3) and revised Eq. (2) into: T=SRs*Rs+Sp*P+c+

All the confusing symbols including Traw, Tadjusted, and 'z' were removed from the revised paper. After revision, the main manuscript and figure captions are consistent. We further revised the figure captions to make them clearer and more concise.

3) Comments: Eq (1), 2 and 5 are just textbook definitions, which are poorly defined for this specific analysis. They use 'a' and 'b' as symbols for different coefficients in 1, 2 and 5. The values for these (a, b) in this analysis may appear in later figures, but the reader has to guess how they were actually computed. Which Figures show which coefficients or adjusted variables is unclear, because they are largely labeled the same: e. g Ts-max or Ta-min, or just 'PC'.

Reply: In this revised paper, we deleted the Eqs. (1) and (3) and revised Eq. (2) (see our response to your last comment). The symbols of 'a', 'b', and 'PC' were removed from the revised paper. Following comments from the other reviewers, the figures of partial correlation coefficients were moved from main text to the supplementary material section with full names labelled.

4) Comments: Relabel PCa, PCb, PCc, PCd etc with a clear connection to a numbered equation coefficient. Use the same specific language to describe the coefficient in both methods and text introducing the Figure.

Reply: See our response to your last comment. We have removed the symbols of 'PC' and relabeled partial correlation coefficients with full names.

5) Comments: Consider adding a simple label to distinguish Tadjusted from T in the Figures.

Reply: In the revised paper, we used 'Adjusted temperatures' (e.g. 'Adjusted Ts-max') instead of Tadjusted.

6) Comments: L177-180 Comment that the number of sunshine duration stations (105 in Wang et al. 2015a) is still small compared with the Ta data. How well are they distributed in western China?

Reply: Wang et al. (2015a) only used the sunshine duration data where direct observations of surface solar radiation were available to make comparison. Sunshine duration and Ta have been observed at each weather station and their numbers are the same. In this study, we used the recently released daily meteorological data at ∼2000 stations, which is the best data one can obtained now. Its spatial distribution was shown in Fig. 1.

7) Comments: L242 What are the coefficients 'a' and 'b'; and their uncertainties? Cross-reference where you show these. When you reach Figs 5 and 6, it is unclear how they relate to Eq (2)

[Figure]

Reply: We revised the equation (see also our response to your comment No. 1). After revision, the main text and figure captions are consistent in the symbols. We have added the 95% confidence intervals to and based on two tailed t-test, e.g. in lines 339-340: "Ts-max was the most sensitive to Rs, followed by Ta-max, and their national means were 0.092 $\pm$0.018 $^\circ$C (W m$-2$)$-1$ (95% confidence) and 0.035$\pm$0.010 $^\circ$C(W m$-2$)$-1$ (95% confidence), respectively." and Lines 374-375: "The national mean sensitivities of Ts-max and Ta-max to P were $-0.321\pm0.098$ $^\circ$C 10 mm$-1$ and $-0.064\pm0.054$ $^\circ$C 10 mm$-1$ (95% confidence), respectively.".

8) Comments: L245 There are no equations 3 and 4.

Reply: The equation (3) was incorrected labelled as the equation (3) in original manuscript. In this revised paper, we deleted the Eqs. (1) and (3), and one equation was kept.

9) Comments: L251 and Figs 2 and 3. Are these Traw or Tadjusted?

Reply: Both Figs 2 and Figs 3 are yearly anomalies of original data of temperatures without adjusting impacts of Rs and P. Only Figs. 7, 8, 9d, and S10 were adjusted temperature and they were explicitly claimed in the figure captions.

10) Comments: Section 3.1.1 and Table 1, all these results are presented as mean trends with no estimate of uncertainty. Add some error estimates.

Reply: We have added the 95% confidence intervals for all of trends in new version.

11) Comments: Section 3.2.1 You need an explicit explanation of Fig 5 and then 6, The reader cannot see clearly how they were constructed. What are these partial correlation coefficients using precipitation as control? Do they relate to the Tadjusted in (5) or the sensitivities in (2)? Nothing has been defined or connected logically (and Eq (3) and (4) are missing? Same issues for Figure 8 and 9.

Reply: We have added an explicit explanation of partial correlation coefficients and the logical connection between partial correlation analysis and multilinear regression

analysis in Methods: "Figs. S3 shows the coefficients of determination of multilinear regression equation (Eq (1)), which indicates how much variance of T can be attributed to that of Rs and P. The high coefficients of determination show that the linear regression performs well, in particular for south China and the North China Plain. To separate the contribution of the Rs and P, we further calculated the partial correlation coefficients between Rs and T (or P and T), which were shown in Figs. S4 and Figs. S5." (Lines 234-240). In addition, we have added explicit introduction in the caption of Figs 5 (Figs S4 in new version): "The linear partial correlation coefficients calculated based on the monthly anomalies of the Rs and T, and avoiding the impact of precipitation (P), which can estimate the proportion of variances of T that can be attributed to the variation of Rs in Eq (1).". We have added similar introduction in the caption of Figs 8 (Figs S5 in new version).

12) Comments: Fig 6 Is this the coefficient 'a' in Eq (2)? Where do you show coefficient 'b'? Is it in Fig9?

Reply: Figs 6 (Figs S7 in new version) show the coefficient 'a' ('$S_{(R_s)}$' in new version) and Figs 9 (Figs S9 in new version) show the coefficient 'b' ('$S_P$' in new version). We have replaced 'a' with '$S_{(R_s)}$' and 'b' with '$S_P$' and used the same symbols in Methods and Figures.

13) Comments: Fig 11 Is this the first time Tadjusted is plotted?

Reply: Yes, it is. We have added the label('adjusted') in all Figures of adjusted temperatures (see Figs 7, Figs 8, Figs 9 and Figs S10 in new version), e.g. 'Adjusted Ts-max' in Figs 7.

14) Comments: L136 and L770 cite different references for the dataset.

Reply: We make it consistent and cited Cao et al.

15) Language issues The structuring of sentences is generally very good, but verbs and tenses need occasional editing, but I will leave this to later editing. An example is

106 LST. . . plays an important role in climate change 107 research because it directly relates to the land surface energy budget. Previously, Ts 108 values used in regional climate research were primarily derived.

Reply: We have carefully checked English usage of this, and tried to make it more concise and clearer. As a result, more than 1400 words was reduced. The manuscript has been sent out for Professional English editing.

---

## Author Comment (AC2) · 20 Jan 2017

Please find our response below your comments

1). General Comment: This paper analyzed the spatial patterns of Ts and Ta and their relations with SSR and precipitation using the observations. It is important to study the mechanism of T changes in the warming climate in regional scales. I think this article is publishable after major corrections.

Reply: Thanks for your high recommendation and the insightful comments, which substantially improve the paper. As a result, four figures were removed and about 1400

words were removed from main text of the revised manuscript. Below please find our point to point response to your comments.

2). Major Comment: Eq (1) is not needed, "linear trend" or "Linear regression" should be enough.

Reply: We have replaced the Eq (1) to a statement in Lines 213-214 that "The linear trends reported in this study were calculated by a linear regression based on monthly anomalies of T, Rs, and P.".

3) Comment: Please discuss why the Tadjusted is calculated? State its actual meaning and applications.

Reply: Thanks for your positive comments. We have added the description in Lines 241-246: "To illustrate the impact of Rs /P on the temperatures, we removed their impacts from their original time series of Ts-max and Ta-max based on multilinear relationship in Eq (1). Then we calculated trends from both the original and adjusted time series. By comparing derived trends of original and adjusted time series, we can quantitatively assess the impact of Rs /P on the Ts-max and Ta-max, in particularly the spatial-temporal pattern of their trends.".

4). Comment: The detailed descriptions are not necessary in Figure captions and can be moved to the text.

Reply: Following the reviewer's suggestion, we substantially reduced the figure captions.

5). Minor Comment: Line 54: Hegerl and Zwiers is missing in the references

Reply: We have added this literature to references. (Lines 523)

6). Comment: Is that 1990?

Reply: Yes, it is 1990. We have changed the 1900 to 1990.

7). Comment: Please check Ts-max and Ta-max trends. By eye, both values should be close.

Reply: We have checked the results of Ts-max and Ta-max trends. The results in paper is right.

8). Comment: Line 277-281: Mechanism of the difference should be mentioned here.

Reply: We have added the mechanism analysis of those difference as followed in main text. "Some previous studies hold an opinion that the microclimate (e.g. urban heat island) has a larger impact on minimum temperatures due to the lower and more stable boundary layer at night (Zhou and Ren, 2011; Christy et al., 2009), while many investigators argued that the variabilities of Rs are the main reason for this daily contrast in warming rates (Sanchez-Lorenzo and Wild, 2012; Makowski et al., 2009)." (Lines 277-281).

9). Comment: Line 284: greater than

Reply: Corrected as suggested.

10). Comment: Line 294: Significant difference. Can you clarify how significant it is please?

Reply: We deleted this sentence from the revised paper. In the revised paper, 95% confidence intervals were added to all the trends.

11). Comment: Line 374: along the coast

Reply: Corrected as suggested.

12). Comment: Line 534: References

Reply: Corrected as suggested.

13). Comment: Line 570: Eastling et al and Line. 638: Ohmura; They are not referenced in the text, please check.

Reply: We cited both references in the revised paper in Lines 274-277: "The warming rate of Ts-min (Ta-min) was significantly faster than that of Ts-max (Ta-max) and the warming rates of all temperatures in cold seasons were substantially stronger than those in warm seasons (Li et al., 2015; Liu et al., 2004; Easterling et al., 1997).".

in Lines 336-342: "Our results are considerably less than the global average dimming rate ($-2.3 \sim -5.1$ W m$-2$ 10yr$-1$) between the 1960s and the 1990s (Gilgen et al., 1998; Liepert, 2002; Stanhill and Cohen, 2001; Ohmura, 2006)."

14). Comment: Ta-min in Figs 5,6,8,9 can be removed, since they don't give much information. They can be briefly discussed in the text.

Reply: Corrected as suggested. We have moved the Figs 5, 6, 8, 9 to the supplementary material and their discussion in main text was substantially reduced.
* * *

---

## Author Response (AR1)

Summary of major revisions: (1) The manuscript was rewritten following the reviewers' suggestions. all the symbols used are consistent. Four figures were removed and about 1800 words were removed from main text of the revised manuscript. (2) The manuscript has been edited by two senior English editors from Nature Springer Language editing services.

**Response to Reviewer #1**

**1.** General Comments**

This is an important paper with interesting Figures on the regional variation of surface temperature trends over China, and their relation to regional precipitation and SSR. The biggest challenge for this reviewer is that I unsure exactly what the elegant Figures show. There are critical gaps between the methods section and the Figures. The legends rather than the text try to explain the content of the Figures, and they are written for the authors, not for a global audience, which will struggle to follow the missing steps in the logic. The reader cannot connect the symbols in Methods to the symbols in the Figures, and the description of the Figures.

**Reply:** Thank you for the high recommendation and constructive comments. We have carefully checked and revised logical structure of paper and unified the symbols for Methods and Figures. As a result, four figures were removed and about 1800 words were removed from main text of the revised manuscript. Below please find our point to point response to your comments.

**2. Technical details**

**Comments:** Methods uses  $T_{raw}$  and  $T_{adjusted}$  and monthly anomalies, as well as 'z' for a regression fit to monthly anomalies of *T*. Do all the graphs show anomalies? Which ones show  $T_{adjusted}$ ? Which ones show regression fits 'z'?

**Reply:** In this study, all of trends and regression analyses are based on the monthly anomalies of temperatures (T, including  $T_{s-max}$ ,  $T_{s-min}$ ,  $T_{a-max}$ ,  $T_{a-min}$ ), surface solar

radiation ( $R_s$ ) and precipitation (P) during 1960-2003. We explicitly claimed in Lines 222-223: "The linear trends reported in this study were calculated via linear regression based on the monthly anomalies of T,  $R_s$ , and P" and in Lines 239-240: "The effect of Rs/P on Ts-max/Ta-max was determined via a multiple linear regression (Roy and Haigh, 2011) of the monthly anomalies using the following equation:".

In this revised paper, we deleted the Eqs. (1) and (3) and revised Eq. (2) into:

$$T = S_{R_s} \cdot R_s + S_P \cdot P + c + \varepsilon$$

All the confusing symbols including  $T_{raw}$ ,  $T_{adjusted}$ , and 'z' were removed from the revised paper. After revision, the main manuscript and figure captions are consistent. We further revised the figure captions to make them clearer and more concise.

**Comments:** Eq (1), 2 and 5 are just textbook definitions, which are poorly defined for this specific analysis. They use 'a' and 'b' as symbols for different coefficients in 1, 2 and 5. The values for these (a, b) in this analysis may appear in later figures, but the reader has to guess how they were actually computed. Which Figures show which coefficients or adjusted variables is unclear, because they are largely labeled the same: e. g Ts-max or Ta-min, or just 'PC'.

**Reply:** In this revised paper, we deleted the Eqs. (1) and (3) and revised Eq. (2) (see our response to your last comment). The symbols of 'a', 'b', and 'PC' were removed from the revised paper. Following comments from the other reviewers, the figures of partial correlation coefficients were moved from main text to the supplementary material section with full names labelled.

**Comments:** Relabel PCa, PCb, PCc, PCd etc with a clear connection to a numbered equation coefficient. Use the same specific language to describe the coefficient in both methods and text introducing the Figure.

**Reply:** See our response to your last comment. We have removed the symbols of 'PC' and relabeled partial correlation coefficients with full names.

**Comments:** Consider adding a simple label to distinguish  $T_{adjusted}$  from *T* in the Figures. **Reply:** In the revised paper, we used 'Adjusted temperatures' (e.g. 'Adjusted Ts-max') instead of  $T_{adjusted}$  (see Figs 7, Figs 8, Figs 9d and Figs S10 in new version).

**Comments:** L177-180 Comment that the number of sunshine duration stations (105 in Wang et al. 2015a) is still small compared with the  $T_a$  data. How well are they distributed in western China?

**Reply:** Wang et al. (2015a) only used the sunshine duration data where direct observations of surface solar radiation are available to make comparison. Sunshine duration and  $T_a$  have been observed at each weather station and their numbers are the same for  $T_a$  and sunshine duration. In this study, we used the recently released daily meteorological data at ~2000 stations, which is the best data one can obtained now. Its spatial distribution was shown in Figure 1.

**Comments:** L242 What are the coefficients 'a' and 'b'; and their uncertainties? Cross-reference where you show these. When you reach Figs 5 and 6, it is unclear how they relate to Eq (2)

**Reply:** We revised the equation (see also our response to your comment No. 1). After revision, the main text and figure captions are consistent in the symbols. We have added the 95% confidence intervals to  $S_{Rs}$  and  $S_P$  based on two tailed t-test, e.g. in lines 342-345: "As shown in Fig S7 shows,  $T_{s-max}$  was the most sensitive to  $R_s$ , followed by  $T_{a-max}$ , and the national means for  $T_{s-max}$  was  $0.092\pm0.018$  °C (W m-2)-1 (95% confidence level) and  $T_{a-max}$  was  $0.035\pm0.010$  °C (W m-2)-1 (95% confidence level)." and Lines 379-381: "The national mean sensitivities of  $T_{s-max}$  and  $T_{a-max}$  to P were  $-0.321\pm0.098$  °C 10 mm-1 and  $-0.064\pm0.054$  °C 10 mm-1 (95% confidence level), respectively.".

**Comments:** L245 There are no equations 3 and 4.

**Reply:** The equation 5 is the third equation in original manuscript. In this revised paper,

we deleted the Eqs. (1) and (3), and one equation was kept.

**Comments:** L251 and Figs 2 and 3. Are these *Traw* or *Tadjusted*?**

**Reply:** Both Fig 2 and Fig 3 are yearly anomalies of original data of temperatures without adjusting impacts of  $R_s$  and P. Only Fig. 7, 8, 9d, and S10 were adjusted temperature and they were explicitly claimed in the figure captions.

**Comments:** Section 3.1.1 and Table 1, all these results are presented as mean trends with no estimate of uncertainty. Add some error estimates.

**Reply:** We have added the 95% confidence intervals for all of trends in new version.

**Comments:** Section 3.2.1 You need an explicit explanation of Fig 5 and then 6, The reader cannot see clearly how they were constructed. What are these partial correlation coefficients using precipitation as control? Do they relate to the  $T_{adjusted}$  in (5) or the sensitivities in (2)? Nothing has been defined or connected logically (and Eq (3) and (4) are missing? Same issues for Figure 8 and 9.

**Reply:** We have added an explicit explanation of partial correlation coefficients and the logical connection between partial correlation analysis and multilinear regression analysis in Methods: "The coefficients of determination ( $R^2$ ) for the multilinear regression equation (Eq (1)) are shown in Fig S3, and they indicate the portion of the variance of *T* that could be attributed to that of  $R_s$  and *P*. High coefficients of determination were obtained, which showed that the linear regression performed well, particularly for South China and the North China Plain. To separate the contributions of  $R_s$  and P, we further calculated the partial correlation coefficients between  $R_s$  and *T* (or *P* and *T*), which are shown in Fig S4 and Fig S5." (Lines 244-250).

In addition, we have added explicit introduction in the caption of Figs 5 (Figs S4 in new version): "The linear partial correlation coefficients calculated based on the monthly anomalies of  $R_s$  and T after avoiding the effect of precipitation (P), which indicates the proportion of variances of T that are attributed to the variation of  $R_s$ .".

We have added similar introduction in the caption of Figs 8 (Figs S5 in new

version).

**Comments:** Fig 6 Is this the coefficient 'a' in Eq (2)? Where do you show coefficient 'b'? Is it in Fig9?

**Reply:** Figs 6 (Figs S7 in new version) show the coefficient '*a*' (' $S_{R_s}$ ' in new version) and Figs 9 (Figs S9 in new version) show the coefficient '*b*' (' $S_P$ ' in new version). We have replaced '*a*' with ' $S_{R_s}$ ' and '*b*' with ' $S_P$ ' and used the same symbols in Methods and Figures.

**Comments:** Fig 11 Is this the first time *Tadjusted* is plotted?

**Reply:** Yes, it is. We have added the label('adjusted') in all Figures of adjusted temperatures (see Fig 7, Fig 8, Fig 9 and Fig S10 in new version), e.g. 'Adjusted  $T_{s-max}$ ' in Fig 7.

Comments: L136 and L770 cite different references for the dataset.

**Reply:** We make it consistent and cited Cao et al.

**3. Language issues**

The structuring of sentences is generally very good, but verbs and tenses need occasional editing, but I will leave this to later editing. An example is 106 LST... plays an important role in climate change 107 research because it directly relates to the land surface energy budget. Previously, Ts 108 values used in regional climate research were primarily derived.

**Reply:** We have carefully checked English usage of this, and tried to make it more concise and clearer. As a result, more than 1400 words was reduced. The manuscript has been edited by two senior English editors from Nature Springer Language editing services.

**Response to Reviewer # 2**

**1. General Comment:**

This paper analyzed the spatial patterns of Ts and Ta and their relations with SSR and precipitation using the observations. It is important to study the mechanism of T changes in the warming climate in regional scales. I think this article is publishable after major corrections.

**Reply:** Thanks for your highly recommendation and the insightful comments, which substantially improve the paper. Below please find our point to point response to your comments.

**2. Major**

**Comment:** Eq (1) is not needed, "linear trend" or "Linear regression" should be enough. **Reply:** We have replaced the Eq (1) to a statement in Lines 222-223 that "The linear trends reported in this study were calculated via linear regression based on the monthly anomalies of *T*,  $R_s$ , and *P*. ".

**Comment:** Please discuss why the  $T_{adjusted}$  is calculated? State its actual meaning and applications.

**Reply:** Thanks for your positive comments. We have added the description in Lines 251-256: "To determine the effect of  $R_s/P$  on the analyzed temperatures, we removed their effects from their original time series of  $T_{s-max}$  and  $T_{a-max}$  based on the multilinear relationship calculated in Eq (1). Then, we calculated the trends from both the original and adjusted time series. By comparing the derived trends of the original and adjusted time series, we quantitatively assessed the effect of  $R_s/P$  on  $T_{s-max}$  and  $T_{a-max}$ , particularly for the spatiotemporal pattern of their trends.".

**Comment:** The detailed descriptions are not necessary in Figure captions and can be moved to the text.

**Reply:** Following the reviewer's suggestion, we substantially reduced the figure

captions.

**3. Minor**

**Comment:** Line 54: Hegerl and Zwiers is missing in the references **Reply:** We have added this literature to references. (Lines 537)

**Comment:** Is that 1990?

**Reply:** Yes, it is 1990. We have changed the 1900 to 1990.

**Comment:** Please check  $T_{s-max}$  and  $T_{a-max}$  trends. By eye, both values should be close. **Reply:** We have checked the results of  $T_{s-max}$  and  $T_{a-max}$  trends. The results in paper is right.

Comment: Line 277-281: Mechanism of the difference should be mentioned here.

**Reply:** We have added the mechanism analysis of those difference as followed in main text. "Although previous studies have indicated that the microclimate (e.g. urban heat island) has a larger effect on minimum temperatures because of the lower and more stable boundary layer at night (Zhou and Ren, 2011; Christy et al., 2009), many investigators argue that variability in  $R_s$  is the primary reason for the daily contrast in warming rates (Sanchez-Lorenzo and Wild, 2012; Makowski et al., 2009)." (Lines 288-292).

**Comment:** Line 284: greater than **Reply:** Corrected as suggested.

**Comment:** Line 294: Significant difference. Can you clarify how significant it is please? **Reply:** We deleted this sentence from the revised paper. In the revised paper, 95% confidence intervals were added to all the trends.

**Comment:** Line 374: along the coast **Reply:** Corrected as suggested.

**Comment:** Line 534: References **Reply:** Corrected as suggested.

**Comment:** Line 570: Eastling et al and Line. 638: Ohmura; They are not referenced in the text, please check.

**Reply:** Both references were cited in the main text. 'Eastling et al' is cited in Lines 285-288: "The warming rate of  $T_{s-min}$  ( $T_{a-min}$ ) was significantly faster than that of  $T_{s-max}$  ( $T_{a-max}$ ) and the warming rates of all temperatures in the cold seasons were substantially greater than those in the warm seasons (Li et al., 2015; Liu et al., 2004; Easterling et al., 1997).". 'Ohmura et al' is cited in Lines 351-354: "Our rate of decrease was considerably less than the global average diminishing rate (form approximately –2.3 to –5.1 W m-2 10yr-1) between the 1960s and the 1990s (Gilgen et al., 1998; Liepert, 2002; Stanhill and Cohen, 2001; Ohmura, 2006)"

**Comment:** Ta-min in Figs 5,6,8,9 can be removed, since they don't give much information. They can be briefly discussed in the text.

**Reply:** Corrected as suggested. We have moved the Figs 5, 6, 8, 9 to the supplementary material and their discussion in main text was substantially reduced.

| 1  | Contributions of Surface Solar Radiation and Precipitation to the Spatiotemporal                            |
|----|-------------------------------------------------------------------------------------------------------------|
| 2  | Patterns of Surface and Air Temperature Warming in China from 1960 to 2003                                  |
| 3  | Jizeng Du 1,2 , Kaicun Wang 1,2* , Jiankai Wang 3 , Qian Ma 1,2 |
| 4  | 1 College of Global Change and Earth System Science, Beijing Normal University,                  |
| 5  | Beijing, 100875, China                                                                                      |
| 6  | 2 Joint Center for Global Change Studies, Beijing 100875, China                                  |
| 7  | 3 Chinese Meteorological Administration, Beijing, 100081, China                                  |
| 8  | Corresponding Authorauthor: Kaicun Wang, College of Global Change and Earth                                 |
| 9  | System Science, Beijing Normal University. Email: kcwang@bnu.edu.cn ; Tel: +086                      |
| 10 | 10-58803143; Fax: +086 10-58800059.                                                                         |
| 11 |                                                                                                             |
| 12 | Submitted to Atmospheric Chemistry and Physics                                                              |
| 13 | November February 168, 20162017                                                                             |
| 14 |                                                                                                             |

**15 Abstract**

| i i |                                                                                                                                                           |
|-----|-----------------------------------------------------------------------------------------------------------------------------------------------------------|
| 16  | Although <del>the</del> global warming has been <del>successfully</del> attributed to <del>the</del>                                                      |
| 17  | elevated increases in atmospheric greenhouses gases, the reasons for mechanisms                                                                           |
| 18  | underlying spatiotemporal patterns the of warming rates trends are still remain.                                                                          |
| 19  | under debate. <del>In this paperHerein, we <del>report</del> analyszed surface and air warming</del>                                               |
| 20  | based on observations recorded collected at 1.977 stations in China from 1960 to                                                                   |
| 21  | 2003. Our results show ed that a significant spatial pattern for the warming of the                                                         |
| 22  | daily maximum surface ( $T_{s-max}$ ) and air ( $T_{a-max}$ ) temperatures showed a significant                                                           |
| 23  | <del>spatial pattern</del> , and the pattern was s tronger in <del>the</del> -northwest China and weaker                                           |
| 24  | in South China and the North China Plain. These warming spatial patterns <del>are</del>                                                                   |
| 25  | were attributed to surface shortwave solar radiation ( $R_{s}$ SSR) and precipitation (P),                                                                |
| 26  | which represent the key parameters of the surface energy budget. During the                                                                               |
| 27  | study period, R s <del>SSR</del> decreased by -1.50 ±0.42 W m -2 10yr -1 in China, and which |
| 28  | caused the trends of in , $T_{s-max}$ and $T_{a-max}$ to decreased by 0.139 and 0.053 °C 10yr -1 ,                                      |
| 29  | respectively. More importantly, the decreasing rates in South China and the North                                                                  |
| 30  | China Plain <del>had an extremelywere much higher <del>dimming rates</del> than those in other</del>                                        |
| 31  | regions. The spatial contrasts of in the trends of $T_{s-max}$ and $T_{a-max}$ in China are were.                                                         |
| 32  | significantly reduced after adjusting for the impact effect of $R_s$ and PSSR and                                                                         |
| 33  | <del>precipitation</del> . For example, after adjusting for the effect of Rs and P, the difference                               |

....

| 34 | in warming rates the T s-max and T a-max values between the North China Plain and |
|----|---------------------------------------------------------------------------------------------------------|
| 35 | the Loess Plateau was reduced by 97.8% and 68.3% for Ts-max and Ta-max,                                 |
| 36 | respectively <del>, After adjusting for the impact of SSR and precipitation,</del> the seasonal  |
| 37 | contrast of in Ts-max and Ta-max decreased by 45.0% and 17.2%, respectively, and                        |
| 38 | the daily contrast of in the warming rates of the surface and air temperature                           |
| 39 | decreased by 33.0% and 29.1% <del>over China, respectively. This study shows showed</del> |
| 40 | that the an essential role of land energy budget in determiningplays an essential                |
| 41 | role in the identification of regional warming patterns,                                                |
|    |                                                                                                         |

**42 **1. Introduction**

I

43 With the rapid development of Increases in observational data and the rapid developments in simulation abilities capacity of climate models have provided evidence 44 45 for the phenomenon of ,-global warming-has been regarded as undeniable (Hartmann et 46 al., 2013), and the increases in anthropogenic greenhouse gases and other 47 anthropogenic impacts effects are believed to beconsidered the primary causes of global warming. However, there are significant spatial and temporal heterogeneities in climate 48 49 warming have been observed, -. i.e. For example, faster warming rates occur in semiarid regions and a "warming hole" has been identified in the central United States (Boyles 50 51 and Raman, 2003; Huang et al., 2012). These spatiotemporal heterogeneities , which

52 represents a major barrier to the reliable detection and attribution of global warming 53 (Tebaldi et al., 2005; Mahlstein and Knutti, 2010). Furthermore, the-uncertainties in 54 model simulations generally increase from the global to the regional scales because of 55 uncertainty in regional climatic responses to global change (Hingray et al., 2007; 56 Mariotti et al., 2011). Therefore, it is crucial to research not only investigations of the 57 spatial and temporal patterns of regional climate changes but alsoand regional climatic 58 response mechanisms to global change are crucial for increasing the accuracy of models 59 designed to detect and explain the causes of. This approach can improve confidence in 60 the detection and attribution of global climate change and predictions of future regional 61 climate change.

62 The spatial heterogeneity of climate warming can be attributed to local climate 63 factors and anthropogenic factors (Karl et al., 1991). For the former local climate factors, 64 local determining factors such as cloud amounts cover and precipitation (P) can significantly influence the speed of regional warming speeds (Hegerl and Zwiers, 2007; 65 66 Lauritsen and Rogers, 2012). Those sSpatial heterogeneities in climate-factor trends make important contributions to have an important influence on various changes in the 67 land-surface energy balance. Existing sStudies have indicated demonstrated that an 68 69 increase in cloud covers can diminishes the surface solar radiation (Rs) downward 70 shortwave solar radiation to the land surface, thusand therefore reducing reduces the

4

daytime temperature (Dai et al., 1997; Zhou et al., 2010; Taylor et al., 2011), although
it has the potential to increase night-time while potentially increasing nighttime
temperatures by intercepting outgoing longwave radiation (Shen et al., 2014; Campbell
and VonderHaar, 1997).

75 Precipitation (P) can alter the proportion of surface absorbed energy partitioned 76 into sensible heat flux-and latent heat flux-: therefore and thereforeit has an 77 inevitable impact effect on both land-surface and near-surface air temperatures (Wang 78 and Dickinson, 2012; Wang and Zhou, 2015). In additionAdditionally, precipitation P 79 plays has a key rolesignificant effect in on the soil thermal inertia and the response of 80 surface vegetation, eausing which results in an important feedback to for regional and 81 global warming (Wang and Dickinson, 2012; Seneviratne et al., 2010; Ait-Mesbah et 82 al., 2015; Shen et al., 2015).

In addition to local climate factors, regional climate systems are significantly affected by the anthropogenic emissions of aerosols-have a significant effect on the regional climate system. Studies have indicated that improving improvements in air quality in recent decades has led to brightening over North America and Europe have led to brightening effect (Wild, 2012; Vautard et al., 2009), whereas surface shortwave solar radiation (SSR) has declined in East Asia and India with increasing air

5

| 89 | pollutionhave led to declines in R s (Xia, 2010; Menon et al., 2002; Wang et al., 2012;   |
|----|------------------------------------------------------------------------------------------------------|
|    |                                                                                                      |
| 90 | Wang et al., 2015a). Consequently, the variations in $\frac{SSR_{Rs}}{Rs}$ may have an impact effect |
| 91 | on both local and global climate change (Wild et al., 2007; Wang and Dickinson, 2013b).              |

I

92 Changes in Land cover change can also alter the energy exchange between the 93 land surface and the atmosphere; moreover, it has and such changes have the potential 94 to impact affect regional climates (Falge et al., 2005; Bounoua et al., 1999; Zhou et al., 95 2004). Previous studies have suggested that urbanization and other land-use changes 96 contribute to promoting the warming effect caused by greenhouse gases (Kalnay and 97 Cai, 2003; Lim et al., 2005; Chen et al., 2015). Overall, the impacts effects of these 98 factors on climate change may be very important on at the regional scale, leading and could lead to a-marked spatial differences in regional climate change, ; whereas 99 100 however, they are usually omitted from the detection and attribution of climate change 101 on-at the global scale (Karoly and Stott, 2006).

102 China has is a vast territory and abundant types that has an abundance of climactic 103 zones stretching from tropical to cold temperate, with and a special alpine climate is 104 observed\_over the Tibet Plateau. In additionAdditionally, the dramatic economic 105 development and explosive population growth in China in recent decades has have 106 caused significant changes in land cover change and serious sever air pollution, **带格式的:** 字体: 倾斜 **带格式的:** 字体: 倾斜, 下标

| 107 | including frequent haze events (Yin et al., 2016; Cheng et al., 2014; Wang et al., 2016). |
|-----|-------------------------------------------------------------------------------------------|
| 108 | The climatic diversity and intensive human activity in this region will likely lead to a  |
| 109 | unique response to global warming with obvious spatial differences in climate change.     |

110 Karl et al. (1991) had analyzed the observational records for the period 1951-1989 111 and, finding found that China's temperature warming trends in China were faster than 112 those of the United States but slower than those of the former Soviet Union. Several 113 studies had have revealed that the warming rate in Northwest China had been was 114 approximately 0.33-0.39 °C 10yr-1 during the second half of the last century (Li et al., 115 2012; Zhang et al., 2010), which was significantly higher than the average warming 116 rate over China (of 0.25 °C 10yr-1) (Ren et al., 2005) or that on a global scale the average 117 global rate of (0.13 °C 10yr-1) (Hegerl and Zwiers, 2007). The Air air temperatures  $(T_{\rho})$ 118 over the Tibet Plateau have has increased by 0.44 °C 10yr-1 over the last 30 years (Duan 119 and Xiao, 2015), which wasand this rate is considerably faster than the overall warming 120 rate in the Northern Hemisphere (0.23 °C 10yr-1) and worldwide (0.16 °C 10yr-1) 121 (Hartmann et al., 2013). To provide insights on global warming and improve the 122 accuracy of future climate change predictions, uUnderstanding the characteristics and 123 mechanisms of regional climate change is critical-to advancing the knowledge and 124 predication of future climate change.

7

| 125 | $T_{a}$ is a common metric for judging determining climate change on the global or             |
|-----|------------------------------------------------------------------------------------------------|
| 126 | regional scales However, I the land surface temperature $(T_{\delta})$ is beginning to play an |
| 127 | increasinglyalso important role in climate change research because of it has the distinct      |
| 128 | advantage of being its directly related relationship to with the land surface energy           |
| 129 | budget. Previously, $T_s$ values used in regional climate research are primarily derived       |
| 130 | from satellite retrievals or reanalysis datasets (Weng et al., 2004; Peng et al., 2014),       |
| 131 | both of which both have good satisfactory global coverage but questionable accuracy            |
| 132 | and integrity. Furthermore, satellite-derived $T_s$ values are only available under clear sky  |
| 133 | conditions, thus limiting their application applicability to in climate change studies.        |
| 134 | In China, both $T_s$ and $T_a$ has been are measured as a conventional meteorological          |
| 135 | observation item parameters by nearly all weather stations, as is $T_a$ . An analysis of the   |
| 136 | spatiotemporal patterns of these parameters identified a close relationship between $T_s$      |
| 137 | and $T_a$ , which indicates that $T_s$ and $T_a$ present equivalent accuracy when used to      |
| 138 | determine This study found that observations of $T_a$ have a good relationship with $T_a$ in   |
| 139 | terms of spatial-temporal patterns and can equally accurately reflect the characteristics      |
| 140 | of climate change. More importantly, $T_s$ is more sensitive than $T_a$ to the local land      |
| 141 | surface energy budget, particularly surface solar radiation (SSR) and precipitation.           |
|     |                                                                                                |

2 From the perspective of energy, bBoth  $\underline{R_s}$  and  $\underline{P}$  SSR and precipitation are key

|----|-------------------------|

| 1 | 带格式的:   | ≤体: 倾斜 |  |  |
|---|----------------|--------|--|--|
|   |                |        |  |  |
|   |                |        |  |  |
|   |                |        |  |  |
|   |                |        |  |  |
|   |                |        |  |  |
|   |                |        |  |  |
| 1 | 带格式的:   | ×体: 倾斜 |  |  |
|   |                |        |  |  |
|   |                |        |  |  |
|   |                |        |  |  |
|   |                |        |  |  |
|   |                |        |  |  |
| 1 | 带格式的: 字 | ≤体: 倾斜 |  |  |
|   |                |        |  |  |
|   |                |        |  |  |
|   |                |        |  |  |
|   |                |        |  |  |
|   |                |        |  |  |
|   |                |        |  |  |
|   |                |        |  |  |
|   |                |        |  |  |
|   |                |        |  |  |
|   |                |        |  |  |
|   |                |        |  |  |
|   |                |        |  |  |
|   |                |        |  |  |

143 factors controlling the land surface energy budget; therefore, their changes in these two 144 factors most likely cause regional differences in the warming rate of  $T_{a}$  (Wild, 2012; Manara et al., 2015; Hartmann et al., 1986). For the first time To our knowledge, this 145 146 study analyzed presents the first analysis of the relationship between Rs (and P) and 147  $T_a/T_s$  between SSR (and precipitation) and  $T_a$  or  $T_s$  in terms of based on their spatial- 148 otemporal patterns and we further quantified the impact effect of the variations of  $R_{\underline{s}}$ 149 and P on  $T_a/T_s$ SSR and precipitation on  $T_a$  and  $T_s$  in China for the period of 1960-150 2003.

151 This paper-article is organized as follows: ... Section 2 introduces the data and 152 methods used in the study. Section 3 includes three parts: the first part describes the 153 spatial and temporal patterns of climate warming over China,: the second part analyzes 154 analyses the impact effect of the variation in  $R_s$  and P on  $T_a/T_{sa}$ SSR and precipitation 155 on  $T_{a}$  and  $T_{a}$ ; and the third part illustrates examines the spatial and temporal patterns of 156 the warming trend of  $T_a/T_s T_a$  and  $T_s$  after adjusting for the impact effects of  $R_s$  and 157 PSSR and precipitation, which eliminated the The adjustment removed impact effects of Rs and P land atmosphereon warming interaction on the warming, leaving impact 158 159 ofand highlighted the effects of large-scale warming caused by the elevated 160 concentrations of atmospheric greenhouse gases-substantially. Moreover, Our results 161 show that adjustment substantially reduced the spatial contrast of in the warming trends

9

162 of  $T_a/T_s$  Ta and Ts in China was substantially reduced after adjusting for the effect of  $R_s$

and P5, and this result is consistent with the expectations under global warming. Finally,

164 Section 4 presents a summary and discussion. which is agree with the expectation of

165 global warming. A summary and discussion are presented in Section 4.

166 2. Data and methods

**167 2.1. Data**

168 The meteorological observational data used in this study are included recently 169 released daily meteorological datasets, including such as the China National Stations' 170 Fundamental Elements Datasets V3.0 (CNSFED V3.0), which can be and they were 171 downloaded from the China's National Meteorological Information Center Centre 172 (http://data.cma.gov.cn/data) (Cao et al., 2016). This These datasets includes included 173 observations of  $T_s$ ,  $T_a$ , the barometric pressure, relative humidity, and sunshine duration. 174 All of the observational records of the climate variables include-were subjected to 175 quality control measures, and homogenization of the processes of data acquisition and 176 compilation.

177 As shown in Figure 1, shows that the number of stations used in this study (1,977 178 selected stations selected from a total of 2,479 stations) is abundant and was ─ **带格式的:** 字体: 倾斜
 ─ **带格式的:** 字体: 倾斜

| 1   |                                                                                                 |  |
|-----|-------------------------------------------------------------------------------------------------|--|
| 179 | significantly greater higher than in that of previous studies (i.e., 57-852 stations) (Kukla    |  |
| 180 | and Karl, 1993; Shen and Varis, 2001; Liu et al., 2004; Li et al., 2015); (Kukla and Karl,      |  |
| 181 | 1993; Shen and Varis, 2001; Liu et al., 2004; Li et al., 2015). therefore Therefore, the        |  |
| 182 | observational data have provided better spatial coverage and higher confidence of in            |  |
| 183 | the detection of detecting regional climate change than in previous studies (Fig. 1). Our       |  |
| 184 | study is the first to use the observations of $T_s$ observation as a parameter for identifying  |  |
| 185 | for research into regional climate change.                                                      |  |
|     |                                                                                                 |  |
| 186 | Observations of $T_s$ at from weather stations are different from $T_s$ data retrieved via      |  |
| 187 | other approaches, such as satellitedataimages and reanalysis. All of the observational          |  |
| 188 | fields of $T_s$ are The $T_s$ observations were performed in 4 - m-× 2 m square bare land plots |  |
| 189 | proximal to the in-a-weather stations. The surface of the observational fields must be          |  |
| 190 | keptwas loose, grassless, and flat, and at the same level as the ground surface of the   |  |
| 191 | weather station. Three thermometers, are placed on the surface of the observational             |  |
| 192 | field, including a surface thermometer, a surface maximum thermometer, and a surface            |  |
| 193 | minimum thermometer were placed. The thermometers are deposited on the surface of               |  |
| 194 | the observational field horizontal to the surface of the observational field, withly: half      |  |
| 195 | of each thermometer is embedded in the soil and the other half is exposed to the air.           |  |
| 196 | When the observational field is was covered by snow, the thermometers are were                  |  |
| 197 | removed from the snow and placed on the snow surface. In additionAdditionally, the              |  |

exposed parts of the thermometers must be were kept cleaned clean to remove from
dust and dew.

To We verifiedy the reliability of the Ts observational records by analyzing, we 200 201 analyzed the the relationship between  $T_a$  and  $T_s$  in the observed records for during 1960–2003. As shown in Figures. S1, the 202 mean Pearson Correlation Coefficients between daily maximum land surface 203 temperature ( $T_{s-max}$ ) and daily maximum air temperature ( $T_{a-max}$ ) calculated from the 204 monthly anomalies were 0.775, 0.843, and 0.806 for the annual, warm, and cold 205 seasonal scales, respectively, and these values were statistically significant (99% 206 confidencelevel) for all stations. The mean correlation coefficients between the daily 207 minimum land surface temperature (Ts-min) and daily minimum air temperature (Tg-min) 208  $T_{s-min}$  and  $T_{a-min}$ -were 0.861, 0.842, and 0.865 for the annual, warm, and cold seasonal 209 scales, respectively, and these values were statistically significant (99% confidence 210 level) for all stations. The high high correlations indicated between Ta and Ts indicates 211 that the observations of either  $T_s$  or  $T_e$  could be used for are reliable for detecting climate 212 change detection.

213 SSR is tThe most fundamental energy resource for  $T_s$  and  $T_a$  is  $R_s$ . In mMost 214 previous studies, had used the observed  $R_s$  have been usedSSR to analyze the 215 relationship between the variation in  $R_s$  SSR and  $T_a$  over Mainland China. However, 一 带格式的: 字体: 倾斜
 一 带格式的: 字体: 倾斜

─ 帶格式的: 字体: 倾斜
 ─ 帶格式的: 字体: 倾斜

|----|---------------------|
|    |                     |
| 4  |                     |

|---------------|-------------------------|
|               |                         |

216 fewer sites were used for  $R_s$ SSR observations than were far less numerous than those 217 for other climatic variables<del>, i.e.; for example</del>, only 85 sites were used for  $R_s$ SSR 218 observations in Liu et al. (2004) and only<del>;</del> 90 sites were used in Li et al. (2015).

More iImportantly, it was found that-sensitivity drifting of the instruments used for the  $R_sSSR$  observations led to a faster dimming rate before 1990, and that instrument replacements from 1990 to 1993 had resulted in a falsely sharp increase in  $SSR_s$  (Wang, 2014; Wagt 2015) The metric that in an analyze the transformation of transformation of transformation of transformation of transformation of transformation of transfor

**223 impeded the wide scientific application of this parameter.**

224 We tTherefore, we used sunshine duration-derived  $\underline{SSRR}_{c}$  in this study, which is 225 based on an effective hybrid model developed by Yang et al. (2006). This model has 226 subsequently been improved (Wang et al., 2015a; Wang, 2014) and it has proved to be 227 performed well in regional and global applications (Tang et al., 2011; Wang et al., 2012). 228 Sunshine duration-derived solar radiation Rs not only can accurately reflects the impact 229 effects of clouds and aerosols on the SSRRs but also ean more exactly reveals long-term 230 SSR trends (Wang et al., 2015a; Wang, 2014). Additionally, Sunshine sunshine 231 duration-derived Rs- values are has a better correlation correlated with the satellite 232 retrievals-derived SSR, reanalysisreanalyzes, and climate model simulations of SSR 233 than the observed SSRRs values observed in Chinafrom observation (Wang et al.,


**234 2015a).**

235 The data are collected by a total ofre are 2,474 meteorological stations reporting data; however, the lengths of the effective observation records for the stations are 236 237 different. In additionAdditionally, only a small number of stations were installed before 238 existed prior to 1960, and the observational records of  $T_s$  at many stations became 239 significantly abnormal-were anomalous after 2003 because of automation. Therefore, 240 in our analysis, we selected 1,977 meteorological stations (see Fig-1) that for which 241 the valid data of observation records with valid data were must be longer than 30 years 242 during the period of 43 years between 1960 and 2003.

243 The monthly anomaly anomalies relative to the 1961-1990 climatology was were 244 calculated based on a monthly mean value of the daily observation-values, and if when 245 a month has was missing more than 7 daily missing values, it that month was classified 246 as a missing value (Sun et al., 2016; Li et al., 2015). The For the annual anomalies, are 247 the average of the monthly anomalies were averaged for the entire year. The anomalies 248 in the warm seasons are-were the averages of the monthly anomalies from May to 249 October, and the anomalies in the cold seasons are were the averages of the monthly 250 anomalies from November to the next April.

A linear regression model (see Eq. (1)) was used to calculate the trend of the

(1)

253 climate variables and can be expressed as:

254

Where where x is time, y is the time series of the monthly anomalies of climate variables, and a and b are the trend and intercept, respectively, regressed using the least-squares method.

258 As shown in Fig-1, the spatial distribution of the weather stations over 259 Mainlandthroughout China is extraordinarily asymmetric and the density of weather 260 stations in East east China is far greater than that in West west China. We used the area-261 weight average method to reduce these biases when calculating the national mean. First, 262 we divided the study region into  $1^{\circ} \times 1^{\circ}$  grids (see Fig. S2) for a total ; there are 953 263 grids covering China. Second, we assigned all selected stations to the grids; there are, 264 and this resulted in 627 grids with containing stations, accounting which accounted for 265 65.79% of the total. Finally, the grid box value is taken to bewas the average of all of the stations on in the grid, and the national mean is was the area-weight average of all 266 of the effective grids (Jones and Moberg, 2003). 267

| 268 | The linear trends reported in this study were calculated byvia a-linear regression         |
|-----|--------------------------------------------------------------------------------------------|
| 269 | based on the monthly anomalies of T, R s , and P. Two national mean trends were |
| 270 | calculated from the anomalies of the grids. In the first method (Method I), the least      |
| 271 | square method. Based on the anomalies of grids, there are two common ways to               |
| 272 | calculate the national mean trends of the variables in China. The first method (Method     |
| 273 | I) calculates the national mean monthly anomalies by were calculated ustaking the area-    |
| 274 | weight of every each grid first, and then calculates the national mean trend based on the  |
| 275 | time series of the national average anomalies was calculated. The In the second method     |
| 276 | (Method II), calculates the trend at every each grid was calculated first, and then the    |
| 277 | national mean trend over China is the area-weighted average value of the was calculated    |
| 278 | from the grid trends on all of the grids.                                                  |

279 In our study, we calculated the national mean trends of the temperatures using 280 Method I and II because both methods as both methods are widely have been used in 281 the existingprevious studies (Gettelman and Fu, 2008). Same The results for the two 282 methods are derived from those two methods if expected to be the same when the time series of all grids is integral-integrated and have no missing data are not missing (Zhou 283 284 et al., 2009): - Howeverhowever, when data are missing, small differences may occur 285 (See Table 1). As shown in Table 1, the absolute value of the difference between Method I and Method II ranged from 0.011 to 0.033 °C 10yr-1, which represented 3.4% to 14.3% 286

| 287                                                  | of the trends (using the results of Method I as the reference). For purposes of                                                                                                                                                                                                                                                                                                                                                                                                                                                                                                                                                                                                                                                                                                                                                                                                                                                                                                                                                                                                                                                                                                                                                                                                                                                                                                                                                                                                                                                                                                                                                                                                                                                                                                                                                                                                                                                                                                                                                                                                                                               |
|------------------------------------------------------|-------------------------------------------------------------------------------------------------------------------------------------------------------------------------------------------------------------------------------------------------------------------------------------------------------------------------------------------------------------------------------------------------------------------------------------------------------------------------------------------------------------------------------------------------------------------------------------------------------------------------------------------------------------------------------------------------------------------------------------------------------------------------------------------------------------------------------------------------------------------------------------------------------------------------------------------------------------------------------------------------------------------------------------------------------------------------------------------------------------------------------------------------------------------------------------------------------------------------------------------------------------------------------------------------------------------------------------------------------------------------------------------------------------------------------------------------------------------------------------------------------------------------------------------------------------------------------------------------------------------------------------------------------------------------------------------------------------------------------------------------------------------------------------------------------------------------------------------------------------------------------------------------------------------------------------------------------------------------------------------------------------------------------------------------------------------------------------------------------------------------------|
| 288                                                  | clarification, the trends derived from Method I are discussed in the main text, whereas                                                                                                                                                                                                                                                                                                                                                                                                                                                                                                                                                                                                                                                                                                                                                                                                                                                                                                                                                                                                                                                                                                                                                                                                                                                                                                                                                                                                                                                                                                                                                                                                                                                                                                                                                                                                                                                                                                                                                                                                                                       |
| 289                                                  | the results from both methods are shown in Table 1.as noted, we selected 1,977 stations                                                                                                                                                                                                                                                                                                                                                                                                                                                                                                                                                                                                                                                                                                                                                                                                                                                                                                                                                                                                                                                                                                                                                                                                                                                                                                                                                                                                                                                                                                                                                                                                                                                                                                                                                                                                                                                                                                                                                                                                                                       |
| 290                                                  | (see Fig. 1) that the valid data of observation records are longer than 30 years during                                                                                                                                                                                                                                                                                                                                                                                                                                                                                                                                                                                                                                                                                                                                                                                                                                                                                                                                                                                                                                                                                                                                                                                                                                                                                                                                                                                                                                                                                                                                                                                                                                                                                                                                                                                                                                                                                                                                                                                                                                       |
| 291                                                  | the period 1960-2003, which is a reasonable compromise between the integrity of the                                                                                                                                                                                                                                                                                                                                                                                                                                                                                                                                                                                                                                                                                                                                                                                                                                                                                                                                                                                                                                                                                                                                                                                                                                                                                                                                                                                                                                                                                                                                                                                                                                                                                                                                                                                                                                                                                                                                                                                                                                           |
| 292                                                  | observation records and the spatial coverage. The missing data in the time series for                                                                                                                                                                                                                                                                                                                                                                                                                                                                                                                                                                                                                                                                                                                                                                                                                                                                                                                                                                                                                                                                                                                                                                                                                                                                                                                                                                                                                                                                                                                                                                                                                                                                                                                                                                                                                                                                                                                                                                                                                                         |
| 293                                                  | some grids results in a little difference between the results of these two methods. To                                                                                                                                                                                                                                                                                                                                                                                                                                                                                                                                                                                                                                                                                                                                                                                                                                                                                                                                                                                                                                                                                                                                                                                                                                                                                                                                                                                                                                                                                                                                                                                                                                                                                                                                                                                                                                                                                                                                                                                                                                        |
| 294                                                  | avoid misunderstanding, the trends derived from Method I was discussed in the main                                                                                                                                                                                                                                                                                                                                                                                                                                                                                                                                                                                                                                                                                                                                                                                                                                                                                                                                                                                                                                                                                                                                                                                                                                                                                                                                                                                                                                                                                                                                                                                                                                                                                                                                                                                                                                                                                                                                                                                                                                            |
| 295                                                  | text, but results from two methods were shown in Table 1.                                                                                                                                                                                                                                                                                                                                                                                                                                                                                                                                                                                                                                                                                                                                                                                                                                                                                                                                                                                                                                                                                                                                                                                                                                                                                                                                                                                                                                                                                                                                                                                                                                                                                                                                                                                                                                                                                                                                                                                                                                                                     |
|                                                      |                                                                                                                                                                                                                                                                                                                                                                                                                                                                                                                                                                                                                                                                                                                                                                                                                                                                                                                                                                                                                                                                                                                                                                                                                                                                                                                                                                                                                                                                                                                                                                                                                                                                                                                                                                                                                                                                                                                                                                                                                                                                                                                               |
| 296
297                                           | Hitdan Hurgsin (a F. A. and the sign has the sign has the sign of the following equation . This can be expressed as:                                                                                                                                                                                                                                                                                                                                                                                                                                                                                                                                                                                                                                                                                                                                                                                                                                                                                                                                                                                                                                                                                                                                                                                                                                                                                                                                                                                                                                                                                                                                                                                                                                                                                                                                                                                                                                                                                                                                                                                                   |
| 296
297
298                                    | Hitdan Hargein (eEQ)-case tech literation by the second s |
| 296
297
298
299                             | Hitdgm Hargei (eEQ) case believe it an hpet flags 65 Repairing the part of flowing equation. This can be expressed as:
$T = S_{R_S} \cdot R_S + S_P \cdot P + c + \varepsilon_{T} = a \cdot x + b \cdot y + c + \varepsilon_{T}$ (21)                                                                                                                                                                                                                                                                                                                                                                                                                                                                                                                                                                                                                                                                                                                                                                                                                                                                                                                                                                                                                                                                                                                                                                                                                                                                                                                                                                                                                                                                                                                                                                                                                                                                                                                                                                                                                                                                                      |
| 296
297
298
299
300                      | Hitdgm Hargei (eEQ) case believe it and put the general sector of the following equation. This can be expressed as:
$T = S_{R_s} \cdot R_s + S_P \cdot P + c + \varepsilon_{\Xi} = a \cdot x + b \cdot y + c + \varepsilon_{\Xi}$ (21)
where $= \underline{T}$ represents the monthly anomalies of $T_{s-max}$ , $T_{s-min}$ , $T_{a-max}$ , and $T_{a-min}$ ; $\underline{S}_{R_s}$ and $\underline{S}_P$                                                                                                                                                                                                                                                                                                                                                                                                                                                                                                                                                                                                                                                                                                                                                                                                                                                                                                                                                                                                                                                                                                                                                                                                                                                                                                                                                                                                                                                                                                                                                                                                                                                                                                              |
| 296
297
298
299
300
301               | Hitdm Hargei (eEQ) as the life if and put the get SP repeited by the part of the following equation. This can be expressed as:
$T = S_{R_S} \cdot R_S + S_P \cdot P + c + \varepsilon_{T} = a \cdot x + b \cdot y + c + \varepsilon_{T}$ (21)
where = T represents the monthly anomalies of T s-max , T s-min , T a-max , and T a-min ; S RS and S P
are the sensitivities of the temperatures to R s and P are the monthly anomalies                                                                                                                                                                                                                                                                                                                                                                                                                                                                                                                                                                                                                                                                                                                                                                                                                                                                                                                                                                                                                                                                                                                                                                                                                                                                                                                                                                                                                                                                                                                                                                                                   |
| 296

302        | Hitdm Hargei (eEQ) case believe if and put flags 6SP repairing the following equation. This can be expressed as:
$T = S_{R_s} \cdot R_s + S_P \cdot P + c + \varepsilon_{\Xi} = a \cdot x + b \cdot y + c + c -$ (21)
where $= T_r$ represents the monthly anomalies of $T_{s-max}$ , $T_{s-min}$ , $T_{a-max}$ , and $T_{a-min}$ ; $S_{Rs}$ and $S_P$
are the sensitivities of the temperatures to $R_s$ and $P_X$ and $y$ are the monthly anomalies
of the SSR and precipitation, respectively; $a$ and $b$ are the corresponding sensitivities                                                                                                                                                                                                                                                                                                                                                                                                                                                                                                                                                                                                                                                                                                                                                                                                                                                                                                                                                                                                                                                                                                                                                                                                                                                                                                                                                                                                                                                                                                                                                                 |
| 296

303 | Hitdyn Hlwgsin (#F.Q) vas tek life eigen hydringer SPR eigen interpreter in the following equation. This can be expressed as:
$T = S_{R_S} \cdot R_S + S_P \cdot P + c + \varepsilon_{T} = a \cdot x + b \cdot y + c + \varepsilon_{-}$ (21)
where = T_represents the monthly anomalies of T_s-max, T_s-min, T_a-max, and T_a-min; S_{R_S} and S_P
are the sensitivities of the temperatures to R_s and P_x and y are the monthly anomalies
of the SSR and precipitation, respectively; a and b are the corresponding sensitivities
of the temperatures to SSR and precipitation, respectively; c is constant term; and c                                                                                                                                                                                                                                                                                                                                                                                                                                                                                                                                                                                                                                                                                                                                                                                                                                                                                                                                                                                                                                                                                                                                                                                                                                                                                                                                                                                                                                                                                      |

| 305 | multilinear regression equation (Eq (1)) are shown in Fig S3, and they indicate the                                           |
|-----|-------------------------------------------------------------------------------------------------------------------------------|
| 306 | portion of the variance of T that could be attributed to that of R s and P. High coefficients                      |
| 307 | of determination were obtained, which showed that the linear regression performed well,                                       |
| 308 | particularly for South China and the North China Plain. To separate the contributions                                         |
| 309 | of $R_s$ and $P$ , we further calculated the partial correlation coefficients between $R_s$ and $T$                           |
| 310 | (or P and T ), which are shown in Fig S4 and Fig S5.                                                            |
| 311 | To adjust determine the effect of $R_s/P$ for the impact of SSR and precipitation on                                          |
| 312 | the analyzed temperatures, we removed their effects from their original time series of                                        |
| 313 | <math>T_{s-max}</math> and <math>T_{a-max}</math> based on the multilinear relationship calculated in Eq (1). Then, we |
| 314 | calculated the trends from both the original and adjusted time series. By comparing the                                       |
| 315 | derived trends of the original and adjusted time series, we quantitatively assessed the                                       |
| 316 | effect of $R_s/P$ on $T_{s-max}$ and $T_{a-max}$ , particularly for the spatiotemporal pattern of their                       |
| 317 | trends.we took x as a time series of SSR and y as a time series of precipitation, while a                                     |
| 318 | and b are the sensitivities of the climate variables to changes in SSR and precipitation,                              |
| 319 | respectively. The method of adjusting for the impact of SSR and precipitation is                                              |
| 320 | expressed as                                                                                                                  |
| 321 |                                                                                                                               |
| 322 | where T adjusted indicates the value of the climate variables after adjusting for the                              |

323 impact of SSR and precipitation and Traw is the value of the climate variables in the raw data.

324 3. Results

325 **3.1. Trends of surface temperature and air temperature**

**326 **3.1.1** The temporal patterns in the variabilities of the temperature variabilitys**

Figs. 2 and Figs. 3 show tThe long-term changes in  $T_{s-max}$  and  $T_{a-max}$  and  $T_{s-min}$  and  $T_{a-min}$  from 1960 to 2003 are shown in Fig 2 and Fig 3, respectively. In addition to the annual variability (Figs. 2a and Figs. 3a), we analyzed the variabilities of the temperature variabilitys in both the warm seasons (May-October) (; Figs. 2b and Figs. 3b) and the cold seasons (November to the following April) (; Figs. 2c and Figs. 3c) were analyzed. In the annual records, all of the temperatures showed exhibited an obvious warming trend over throughout China (Figs. 2a and Figs. 3a).

As shown in Table 1, the national mean warming rate from 1960 to 2003 for  $T_{s-max}$

was  $0.227 \pm 0.091$  °C  $10 \text{ yr}^{-1}$  (95% confidence level) and the rate for  $T_{a-max}$  was

 $0.167 \pm 0.068$  °C 10yr-1 (95% confidence level) from 1960 to 2003. The warming rate of

B37 Ta-max based on the 1,977 stations examined in this paper the current study was a

338 littleslightly higher than both that of the global average (0.141 °C 10yr-1) from 1950 to

339 2004 (Vose et al., 2005) and that the rate obtained from of a previous analysis of China

19

|----|---------------------|


341 (Liu et al., 2004). Additionally, the increases in

340

The seasonal contrasts of warming of  $T_{a-max}$  and  $T_{s-max}$  are important.  $T_{s-max}$  had an average rate of 0.172 °C 10yr-1 in the warm seasons and 0.354 °C 10yr-1 in the cold seasons. For  $T_{a-max}$ , it was 0.091 °C 10yr-1 and 0.294 °C 10yr-1 in the warm and cold seasons, respectively. The increases in  $T_{s-max}$  and  $T_{a-max}$  in the cold seasons were much

(0.127 °C 10yr-1) of temperatures from 1955 to 2000 based on 305 stations in China

larger than those in the warm seasons, which is consistent with previous studies ofChina and other regions (Shen et al., 2014; Vose et al., 2005; Ren et al., 2005).

Similarly, the warming rates of  $T_{s-min}$  and  $T_{a-min}$  in the warm seasons were clearly

also clearly lower than those in the cold seasons too. As shown in Fig 3, *Ts-min* increased

by  $0.315\pm0.058$  °C  $10yr^{-1}$  (95% confidence level) and  $T_{a-min}$  increased by

351 0.356±0.0057 °C 10yr-1 (95% confidence level) (see Fig 3a) from 1960 to 2003.As

shown in Figs. 3, *Ts-min* increased by 0.315 °C 10yr-1 and *Ta-min* increased by 0.356 °C

353 10yr-1 (see Figs. 3a) from 1960 to 2003. The warming trend of  $T_{a-min}$  is generally

354 consistent with earlier studies (Shen et al., 2014; Li et al., 2015; Liu et al., 2004);

bowever, it-these trends is are considerably larger than that the rates reported for the

global average (0.204 °C 10yr-1) (Vose et al., 2005). For the seasonal scales, the

357 warming rate of  $T_{s-min}/T_{a-min}$  increased at a rate of 0.221 °C 10yr-1 in the warm seasons

and 0.447 °C 10yr-1-in the cold seasons from was almost double that of the warm
 seasons from 1960 to 2003 (see Table 1). Ta-min increased at rates of 0.245 °C 10yr-1
 and 0.505 °C 10yr-1 in the warm and cold seasons, respectively.

361 On a national average scale, all temperatures increased from 1960 to 2003. The 362 warming rate of  $T_{s-min}$  ( $T_{a-min}$ ) was significantly faster than that of  $T_{s-max}$  ( $T_{a-max}$ ) and the 363 warming rates of all temperatures in the cold seasons were generally substantially 364 higher greater than those in the warm seasons. These basic characteristics of the 365 temperature changes are consistent with previous studies on global or regional scales 366 (Li et al., 2015; Liu et al., 2004; Easterling et al., 1997). (Hartmann et al., 2013). 367 Although previous studies have indicated that the microclimate (e.g. urban heat island) 368 has a larger effect on minimum temperatures because of the lower and more stable 369 boundary layer at night (Zhou and Ren, 2011; Christy et al., 2009), many investigators 370 argue that variability in  $R_s$  is the primary reason for the daily contrast in warming rates 371 (Sanchez-Lorenzo and Wild, 2012; Makowski et al., 2009). (Liu et al., 2004; Karl et al., 372 1993)However, there remain slight differences between our results and previous studies 373 with respect to the temperature warming rates, which might have several causes. 374 The number of stations used in our study is much greater in previous studies, which 375 has led to better spatial coverage and a better representation of our analytical result. In

**377 **3.1.2. The sS**patial patterns in the variabilities for the temperature variabilitys**

As shown in Figs. 4, demonstrates a clear spatial heterogeneity was demonstrated in the warming

 $\frac{1}{380}$  at high rate and the trends of  $T_{s-max}$  and  $T_{a-max}$  were statistically significant higher in for

rates for  $T_{s-max}$  and  $T_{a-max}$  over in China from 1960-to 2003. Transmit and  $T_{a-max}$  increased

the Tibet Plateau, and Northwest and Northeast China (see Figs S36). However, Ts max

382 and Ta max had a relative lower warming rate in the compared with the North China Plain

and South China, and Ts and the showed cooling Cooling trends in Ts and the second sec

384 detected for the Sichuan PlainBasin, the Yangtze River Delta, and the Pearl River Delta.

Lower warming rates of warming of  $T_{\mu-max}$  in South China and the North China Plain

386 had have also been previously reported in multiple previous studies (Liu et al., 2004;

387 Li et al., 2015).

376

379

For The warming rates of  $T_{s-max}$  and  $T_{a-max}$ ; the warming rates of in South China and the North China Plain in the warm seasons were considerably lower than those in the cold seasons, resulting which resulted in a more obviousstronger spatial heterogeneity in the warm seasons (Figs. 4b and 4h). However, the warming rates of both  $T_{s-max}$  and  $T_{a-max}$  in the Sichuan Basin and the Pearl River Delta were lower in the cold seasons than in the warm seasons. Despite of T the spatial and seasonal patterns of

|----|---------------------|

|----|-------------------------|

|---------------------|
|
                |

 $T_{a-max}$  were similar, although they were not as elearly similar to as the those patterns. 395 of  $T_{s-max}$ . The spatial contrast in the trends between For  $T_{a-max}$  both the seasonal *asymmetry and the spatial heterogeneity of the warming trend were less than those of*  $T_{s-max}$ -

| 398 | For $T_{s-min}$ and $T_{a-min}$ was much less than that between $T_{s-max}$ and $T_{a-max}$ , although                    |
|-----|---------------------------------------------------------------------------------------------------------------------------|
| 399 | a strong dependence on latitude was observed the warming rates were highest in North                                      |
| 400 | China and generally decreased from north to south (Figs. 4d and 4j). The average                                          |
| 401 | warming rates of $T_{s-min}$ and $T_{a-min}$ in the cold seasons (Figs. 4f and 4l) were faster than                       |
| 402 | those in the warm seasons (Figs. 4e and 4k). This variation of warming rate with                                          |
| 403 | latitudes have This dependence has been successfully -been-attributed to amplified                                        |
| 404 | dynamics amplification (Wallace et al., 2012; Ding et al., 2014). In this study, we focus                                 |
| 405 | on the spatial heterogeneity of the warming rates at similar latitudes and diurnal contrast                               |
| 406 | of the warming rates.                                                                                                     |
| 407 | By contrasting the annual variation and spatial pattern of trends, we found that The                                      |
| 408 | correlation between <math>T_s</math> and <math>T_a</math> was highly had an extremely significant correlation with |

409 each other. Based on the time series of the national mean yearly anomalies (see Figs. 2

410 and Figs. 3), the correlation coefficients between  $T_{s-max}$  and  $T_{a-max}$  were was 0.877\_

411 0.799, and 0.921 on the annual, warm, and cold seasonal scales, respectively. The

|----|---------------------|

| 412 | correlations-and between Ts-min and Ta-min were-was_0.976_, 0.969, and 0.977-on the                                                |           |
|-----|------------------------------------------------------------------------------------------------------------------------------------|-----------|
| 413 | annual, warm, and cold seasonal scale. s, respectively. In the spatial pattern of the                                              |           |
| 414 | trendsIn the spatial pattern of the trends (Figs. 4), the correlation coefficients between                                         |           |
| 415 | $T_{s-max}$ and $T_{a-max}$ were was 0.488 and , 0.465, and 0.522 on the annual, warm, and cold                                    | _         |
| 416 | seasonal scales, respectively. Those-between $T_{s-min}$ and $T_{a-min}$ were-was 0.638, 0.670,                                    | _         |
| 417 | and 0.594-on the annual, warm, and cold seasonal scales, respectively. All of these                                                |           |
| 418 | correlations between $T_{\delta}$ and $T_{a}$ were significant at the 95% significance level, which                                |           |
| 419 | indicated a close relation between $T_s$ and $T_a$ for both interannual fluctuations and secular                                   |           |
| 420 | trends                                                                                                                             |           |
|     |                                                                                                                                    |           |
| 421 | In summary, T s -had a significant correlation with Ta both in annual variation (Figs.                                  |           |
| 422 | 2 and Figs. 3) and in long-term trends (Figs. 4), indicating that T s observational records                             |           |
| 423 | are reliable for climate change research. However, $t_{\underline{T}}$ he correlation between $\underline{T}_{s-min}$ and          | /         |
| 424 | $T_{a-min}$ was significantly higher than that between $T_{s-max}$ and $T_{a-max}$ . $T_{s-min}$ is closely related                | $\langle$ |
| 425 | to the landatmosphere longwave wave radiation balance during the nighttime at night,                                               |           |
| 426 | which is closely related associated to with the atmospheric greenhouse effect (Dai et al.,                                         |           |
| 427 | 1999). During the day time, $T_s$ is directly determined by the land surface energy balance,                                       | /         |
| 428 | i.e., the incoming energy (including $SSRR_{s}$ -) and atmospheric longwave radiation                                              |           |
| 429 | (Wang and Dickinson, 2013a), and it is partitions-partitioned into latent and sensible                                             |           |
| 430 | heat fluxes (Zhou and Wang, 2016). Despite Although its Ta is dependence dependent on the land-atmosphere | _         |

|----|---------------------|

|-------------------|----------------------|
|                   |                      |
| _                 | 带怒子的 • 之休· 倾斜 |
| 1                 |                      |
| $\langle \rangle$ |                      |

|---|---------------------|
|   |                     |
|   |                     |

|----|---------------------|
| -( |                     |

| 431 | sensible heat flux, it <math>T_{\alpha}</math> is also impacted affected by local and/or large-scale circulation. | / |
|-----|--------------------------------------------------------------------------------------------------------------------------|---|
| 432 | So Thus, the changes of in the land surface energy balance caused by $SSR_{R_{\delta}}$ and                              | / |
| 433 | precipitation P have different levels of effect on $T_s$ and $T_a$ during the day, which most       | < |
| 434 | likely causes caused a the lower correlation between $T_{s-max}$ and $T_{a-max}$ than that between                | < |
| 435 | $T_{s-min}$ and $T_{a-min}$ .                                                                                            | < |
| 436 | 3.2. <del>The impact ofEffect of surface solar radiationRs and precipitation_P on</del>  | < |
| 437 | temperatures                                                                                                             |   |

**438 **3.2.1 Effect of** *R*s**Impact of surface solar radiation**

439 -As shown in Figs. S4, shows that  $SSR_{R_s}$  had is closely an important

440 relationshiplinked with  $T_{s-max}$  and  $T_{a-max}$  but not with  $T_{s-min}$  and  $T_{a-min_3}$  and the correlation

441 and  $T_{a-min}$  between  $T_{s-max}$ - and  $R_s$  was higher than that between  $T_{a-max}$  and  $R_s$  The national

442 mean of the partial correlation coefficients between SSR and Ts-max is 0.552 and 98.9%

443 of the stations are statistically significant at the 1% level. Meanwhile, the national mean

44 of the partial correlation coefficients between SSR and Ta-max is 0.441, and 95.4% of

the stations are statistically significant at the 1% level. This relationship is stronger in

446 South China and on the North China Plain, i.e., it reaches 0.810 for Ts-max and 0.765 for

447 <del>Ta-max. –</del>

|----|---------------------|---|
|    |                     |   |
| (  | #**                 |   |

| 448 | Ţ indo alladingub kel i <mark>tudi kalingi in kini di kukan politi labali ter (1884) date eli 1865 alla alla di ter (11168), je</mark> |   |                    |            |
|-----|-----------------------------------------------------------------------------------------------------------------------------------------------|---|--------------------|------------|
| 449 | warm seasons is was higher than that in the cold seasons, and this correlation was                                                            |   |                    |            |
| 450 | stronger in South China and the North China Plain, the national mean partial correlation                                                      |   |                    |            |
| 451 | coefficients for the warm and cold seasons are 0.579 and 0.498 for $T_{s-max}$ and 0.544 and                                                  |   |                    |            |
| 452 | $0.386$ for $T_{a-max}$ , respectively, consisting with the seasonal cycle of SSR intensity over                                              |   |                    |            |
| 453 | China.                                                                                                                                        |   |                    |            |
|     |                                                                                                                                               |   |                    |            |
| 454 | Spatially, overall, the partial correlation coefficients between $T_{s-max}$ and $T_{a-max}$ and                                              |   |                    |            |
| 455 | SSR are higher in South China than in North China (see Figs. 5a 5c and 5g 5i). South                                                          |   |                    |            |
| 456 | of 35° N, the national mean of the partial correlation coefficients between T s-max (T a-max )                          |   |                    |            |
| 457 | and SSR is 0.654 (0.552), whereas that between $T_{s-max}$ ( $T_{a-max}$ ) and SSR is just 0.417                                              |   |                    |            |
| 458 | (Shen et al., 2014) north of 35° N. During daytime, $T_s$ and $T_a$ is largely determined by                                                  |   |                    |            |
| 459 | how much energy is used to evapotranspiration. SIn south China has highwhere soil                                                             |   |                    |            |
| 460 | moisture-is-high,; therefore, the relationship between the energy used for                                                                    |   |                    |            |
| 461 | evapotranspiration and is near linearly related to SSRRs is approximately linear (Wang                                                        | / | 带格式的:下             | 标          |
| 462 | and Dickinson, 2013b; Zhou et al., 2007). However, northwest China presents dry soil                                                          |   |                    |            |
| 463 | over most of the year; thus the energy used for evapotranspiration is more dependent                                                          |   |                    |            |
| 464 | on precipitation in the northwest China where the soil is dry during most time of a                                                           |   | 带格式的:字             | 体: 倾斜      |
| 465 | yearP. As a result, the energy available for heating the surface and air temperatures is                                                      |   |                    |            |
| 466 | not asso closely correlated with SSRRs. Therefore, the correlation coefficients between                                                       | _ | 带格式的: 字
标 |

| 468 | To quantify the impact effect of $SSR_{\underline{R}_{\underline{\delta}}}$ on temperature, the sensitivity of the                           | /         |
|-----|----------------------------------------------------------------------------------------------------------------------------------------------|-----------|
| 469 | studied temperatures to changes in SSRRs has been was calculated (Eq. (21)). As shown                        |           |
| 470 | in Figs. 6Fig S7 shows, $T_{s-max}$ was the most sensitive to SSR $R_s$ , followed by $T_{a-max}$ , and                                      | $\langle$ |
| 471 | their national means were for $T_{s-max}$ was 0.092±0.018 °C (W m -2 ) -1 (95% confidence                              | <         |
| 472 | level ) and <math>T_{a-max}</math> was 0.035±0.010 °C (W m -2 ) -1 (95% confidence level), respectively. |           |
| 473 | $T_{s-min}$ and $T_{a-min}$ were insignificantly not sensitive to SSR $R_s$ because these temperatures                                       | <         |
| 474 | are primarily affected by they primarily depend on atmospheric longwave radiation                                                            |           |
| 475 | during the nighttimenight.                                                                                                                   |           |
| 476 | Based on the above analysis, we calculated the impact effect of changes in $\frac{SSR_{x}}{R_{x}}$                                           | /         |
| 477 | on the studied temperatures (see the Method Section). From 1960 to -2003, the                                                                |           |
| 478 | calculations of the monthly anomalies at 1,977 stations indicated that the national mean                                                     |           |
| 479 | rate of decreasing ratee of SSRR was $-1.502\pm0.42$ W m -2 10yr -1 (95% confidence)                     |           |
| 480 | level), as calculated from monthly anomalies at 1,977 stations, and the trend was                                                            |           |
| 481 | significant in most regions over of China (see Figs. S4Fig S8) Our results rate of                                                           |           |
| 482 | decrease was are considerably less than the global average dimming diminishing rate                                                          |           |
| 483 | ( form approximately $-2.3 - to -5.1 \text{ W m}^{-2} 10 \text{ yr}^{-1}$ ) between the 1960s and the 1990s                           |           |
| 484 | (Gilgen et al., 1998; Liepert, 2002; Stanhill and Cohen, 2001; Ohmura, 2006) and the                                                         |           |

|--------|-------|----------|
|        |       |          |
|        |       |          |
|        |       |          |

|----|---------------------|

national mean dimming rate across China (from approximately -2.9 to- -5.2 W m-2
10yr-1) between the 1960s and the 2000s based on radiation station observations (Che
et al., 2005; Liang and Xia, 2005; Shi et al., 2008; Wang et al., 2015a).

488 As noted in the data section, the sensitivity drifting and replacement of the 489 instruments used for the SSRRs observations results resulted in a significant 490 homogenization in of the stations observation records (Wang, 2014; Wang et al., 2015a), 491 which eauses introduced considerable a great uncertainty in to the trend\_estimations. 492 Tang et al. (2011) used quality-controlled observational data from 72 stations and two 493 radiation models based on 479 stations to determine both that the dimming rate inover China is decreased from approximately -2.1 -to -2.3 W m-2 10yr-1 during 1961-2000, 494 495 and that thethey also showed that SSRRs values has have remained been essentially 496 unchanged since 2000.; this These findings is are generally consistent with our results. 497 Due to Because of the decreasing trend in  $\frac{SSR_{R_s}}{R_s}$ , the national mean warming 498 trends of Ts-max and Ta-max decreased by 0.139 °C 10yr-1 and 0.053 °C 10yr-1 respectively, in the national mean. Spatially, the decreasing rate of SSRRs in South 499 500 China and the North China Plain was significantly higher than that in other regions,

501 especially particularly in the warm seasons (Figs. 7Fig 5b). Therefore, the cooling effect

of decreasing  $\frac{SSR_{R_s}}{r_s}$  on  $T_{s-max}$  and  $T_{a-max}$  was more significant in South China and the

| 6 | -     |        |
|---|-------|--------|
|   |       |        |

|--------|---------------------|
| $\neg$ |                     |
| 503 | China North Plain, and it resulted ing in significantly lower warming rates of $T_{s-max}$ and        |  |
|-----|-------------------------------------------------------------------------------------------------------|--|
| 504 | $T_{a-max}$ in those regions there—than in the other regions (see Figs. 4). The spatial               |  |
| 505 | consistency between the decreasing $\frac{SSRR_s}{r_s}$ trend and the warming slowdown of $T_{s-max}$ |  |
| 506 | $(\underline{T}_{a-max})$ implies warming implied that variations in SSRRs is were the primary reason |  |
| 507 | for the spatial heterogeneity of the warming rate in $T_{s-max}$ ( $T_{a-max}$ ).                     |  |

**508 3.2.2 Effect of P Impact of Precipitation**

| 509 | As shown in Fig S5, Figs. 8a a shows that there is a significant negative correlation                     |
|-----|------------------------------------------------------------------------------------------------------------------|
| 510 | was detected between Ts-max and precipitationP , and; the national correlation was more |
| 511 | significant in the warm seasons than in the cold seasons. Pmean of the partial                                   |
| 512 | correlation coefficients is -0.323, and 99.3% of the stations are statistically significant                      |
| 513 | at the 1% level. Seasonally, the correlation is stronger in the warm seasons (regional                           |
| 514 | mean: -0.405) than in the cold seasons (regional mean: -0.276). In warm seasons, the                             |
| 515 | correlation in North China (regional mean: -0.459) is clearly stronger than in South                             |
| 516 | China (regional mean: -0.365). In cold seasons, the correlation is highest on the                                |
| 517 | Southwestern Yunnan-Guizhou Plateau and in most regions of North China (regional                                 |
| 518 | mean: -0.305) (Figs. 8b and 8c), whereas it was is relatively weak in Southeastern                               |
| 519 | China, the Tibet Plateau, Dzungaria, the Tarim Basin, and some regions of Northeastern                           |
| 520 | China (regional mean: -0.117). The correlations between T a max and precipitation had                 |
|     |                                                                                                                  |

|---------------------|
|                     |
|                     |

|----|---------------------|

have similar spatial and seasonal patterns (Figs. 8g 8i) too, and 35.4% of the stations
had a correlation between Ta-max and the precipitation that wasare statistically
significant at the 1% level; these were are primarily concentrated in arid and semiarid
regions of China (regional mean: -0.167) (Figs. 8e 8f and 8j 8l).

525Precipitation has a negatively relationship correlated with temperature because526precipitation P can reduces temperatures by increasing the surface evaporative cooling527(Dai et al., 1997; Wang et al., 2006). The impact of precipitation on temperature was528is higher in the warm seasons over China, which is consistent with seasonal changes in529the correlation between  $T_{s-max}$  and  $T_{a-max}$  and precipitation (see Figs. 8b - 8e and 8h - 8i). T

| 530 | The national mean sensitivities of $T_{s-max}$ and $T_{a-max}$ to precipitation were                 |
|-----|-------------------------------------------------------------------------------------------------------------|
| 531 | $-0.321\pm0.098$ °C 10 mm -1 and $-0.064\pm0.054$ °C 10 mm -1 (95% confidence level), |
| 532 | respectively. As shown in Figs. 9Fig S9, there were apparent seasonal and spatial                           |
| 533 | changes in the sensitivity of $T_{s-max}$ and $T_{a-max}$ to precipitation P were apparent (Figs.    |
| 534 | 9 Fig S9 a–9c and Fig S9 9g–9i). In warm seasons, these sensitivities were highest in the     |
| 535 | Tibet Plateau, the Loess Plateau, the Inter Mongolia Plateau, Dzungaria, and the Tarim                      |
| 536 | Basin (Figs. 9b and 9h). In cold seasons, the distribution of regions with high sensitivity                 |
| 537 | extended to all of North China and Southwest China (Figs. 9c and 9i). Overall, tThe                         |
| 538 | sensitivities of $T_{s-max}$ were significantly higher in arid regions (dry seasons)                        |

|---|---------------------|
| 1 |                     |

|----|---------------------|

|---|---------------------|

| 539 | than in-humidity regions (rainy seasons) (Wang and Dickinson, 2013b). In contrastAs                          |                                                                   |
|-----|--------------------------------------------------------------------------------------------------------------|-------------------------------------------------------------------|
| 540 | expected, $T_{s-min}$ and $T_{a-min}$ were both less sensitive to variations in the precipitation P . | 带格式的: 字体: 倾斜
| 541 | As Figs. 10 shows, during 1960-2003, tThe trend in the precipitation P from 1960_                            | 带格式的: 字体: 倾斜                                               |
| 542 | to 2003 over the 1.977 stations had showed obvious spatial heterogeneities. China's                          |                                                                   |
| 543 | precipitation during this period showed a A slight increasing trend in P was observed in                     | 带格式的: 字体: 倾斜                                               |
| 544 | China during this period at with an increasing-rate of 0.112±0.718 mm 10yr -1 _(95%)              |                                                                   |
| 545 | confidence level). An increasing Precipitation P trend was observed in Northwestern                          | 带格式的: 字体: 倾斜                                               |
| 546 | northwestern_China and Southeastern_southeastern_China-experienced an increasing                             |                                                                   |
| 547 | trend, whereas a decreasing trend was observed in the precipitation in the North China                       |                                                                   |
| 548 | Plain, the Sichuan Basin, and parts of Northeastern-northeastern China-experienced a                         |                                                                   |
| 549 | decreasing trend. However, the trend of precipitation P trends was were not insignificant             | 带格式的: 字体: 倾斜                                               |
| 550 | in most regions (see Figs. S4Fig S8). Variations in precipitation Phad significantly                         | 带格式的: 字体: 倾斜                                               |
| 551 | differed by seasonal differences (see Figs. 10Fig 6b and Fig 610c). The seasonal and                         |                                                                   |
| 552 | spatial characteristics variations in of these precipitation variations P are consistent with                | 带格式的: 字体: 倾斜                                               |
| 553 | those identified inof previous studies (Zhai et al., 2005; Wang et al., 2015b).                              |                                                                   |
| 554 | Therefore, $f\underline{F}$ or $T_{a-max}$ and $T_{s-max}$ , the reduction in precipitation aggravated the   |                                                                   |
| 555 | warming trend in the North China Plain, the Sichuan Basin, and parts of Northeastern                         |                                                                   |
| 556 | northeastern China was aggravated by the reduction in P , whereas the warming trend                   | 带格式的: 字体: 倾斜                                               |

|---|---------------------|
|   |                     |

|---|--------------------|
|   |                    |
|   |                    |
|   |                    |
|   |                    |
|   |                    |
|   |                    |
|   |                    |
|   |                    |
|   |                    |
|   |                    |
|   |                    |
|   |                    |
| 6 | NU FAL IN FF       |

**的:** 字体: 倾斜

| 557        | increase in precipitation primarily slowed the warming trend in Nnorthwestern China                                                          |   |                                                                          |
|------------|----------------------------------------------------------------------------------------------------------------------------------------------|---|--------------------------------------------------------------------------|
| 558        | and on in the Mongolian Plateau were slowed by increases in P (Figs. 10Fig 6 d). On                                                   |   | 带格式的: 字体: 倾斜                                                      |
| 559        | For the national average, the impact effect of increasing precipitation P resulted in                                                        |   | 带格式的: 字体: 倾斜                                                      |
| 560        | decreases in the warming trends of $T_{s-max}$ and $T_{a-max}$ being decreased by -0.007 °C                                                  |   |  <li>( 带格式的: 字体: 倾斜</li> <li>( 带格式的: 字体: 倾斜</li>  |
| 561        | 10yr -1 and -0.002 °C 10yr -1 , respectively. However, compared to SSR, the impact                                     |   |                                                                          |
| 562        | effect of precipitation P on $T_{s-max}$ was smaller by approximately an order of magnitude                                    | _ |  <li>【 带格式的: 字体: 倾斜</li> <li>【 带格式的: 字体: 倾斜</li>                |
| 563        | less than that of $R_{\underline{s}}$ . For $T_{\underline{s}$ -min- and $T_{\underline{a}$ -min, the impact of changes in precipitation was |   |                                                                          |
| 564        | insignificant.                                                                                                                               |   |                                                                          |
| 565
566 | 3.3. Trends of surface and air temperature after adjusting for the effect of SSRR s and precipitationP                         | / | ( 带格式的: 下标
( 带格式的: 字体: 倾斜                                             |
| 567        | Based on the above analysis of the impact effect of SSRR and precipitation P on                              | _ |  <li>【 带格式的: 下标</li> <li>【 带格式的: 字体: 倾斜</li>                    |
| 568        | temperatures, we found that the variations of in SSRR and precipitation P had little                                           | _ |  <li>( 带格式的: 下标</li> <li>( 带格式的: 字体: 倾斜</li>                    |
| 569        | effect on $T_{s-min}$ and $T_{a-min}$ . However, $R_s$ and $P$ had important effect on the trends of $T_{s-}$                                | _ |  <li>( 带格式的: 字体: 倾斜</li> <li>( 带格式的: 字体: 倾斜</li>                |
| 571        | closely related to $R_s$ (see Fig S4). Therefore Therefore, we only the effects of $R_s$ and $P$                                             |   |                                                                          |
| 572        | on $T_{s-max}$ and $T_{a-max}$ were analyzed analyzed their impact on $T_{s-max}$ and $T_{a-max}$ . After                                    |   |                                                                          |
| 573        | adjusting for the impact effect of SSRRs and precipitation P (Figs. 11Fig 7), the                                                            |   | (带格式的: 字体: 倾斜                                                            |
| 574        | warming rates of $T_{s-max}$ and $T_{a-max}$ increased by 0.146 °C 10yr -1 (64.3%) and 0.055 °C                                   |   | 带格式的: 字体: 倾斜
|            |                                                                                                                                              |   |                                                                          |

**575 10yr-1 (33.0%), respectively.**

| 576 | After adjustingAdditionally, the increasing amplitude of warming rates in the                                            |
|-----|--------------------------------------------------------------------------------------------------------------------------|
| 577 | warm seasons was significantly higher than that in the cold seasons, which resulted in                                   |
| 578 | the a seasonal contrast in warming rates, with of $T_{s-max}$ and $T_{a-max}$ decreasinged by 45.0% |
| 579 | and 17.2% respectively (see Table 1). The national mean warming rate of T s-max                               |
| 580 | increased by 0.178 °C 10yr -1 (103.1%) in the warm seasons and 0.086 °C 10yr -1 (27.2%)            |
| 581 | in the cold seasons. For T a-max , the warming rate increased by 0.069 °C 10yr -1 (76.4%)          |
| 582 | in the warm seasons and 0.034 °C 10yr -1 (11.7%) in the cold seasons                                          |
|     |                                                                                                                          |
| 583 | After adjusting for the impact of SSR and precipitation , the difference in warming                               |
| 584 | rates between $T_{a-max}$ and $T_{a-min}$ changed from 0.190 to 0.134 °C 10yr -1 , a decrease of              |
| 585 | 29.1%, and the difference between $T_{s\mbox{-max}}$ and $T_{s\mbox{-min}}$ changed from 0.088 to 0.058 °C               |
| 586 | <del>10yr-1, a decrease of 33.0%.</del>                                                                       |
|     |                                                                                                                          |
| 587 | More importantly, after adjusting for the impact effect of $SSRR_{s}$ and                                         |
| 588 | precipitation P , the spatial coherence of the warming rates of $T_{s-max}$ and $T_{a-max}$ in South       |
| 589 | China and the North China Plain clearly improved (Figs. 12Fig 8). The regional                                           |
| 590 | differences between among the North China Plain, South China, and other regions in                                       |
| 591 | China shrank significantly due to decreased because of the increase in the warming rates                                 |
| 592 | in South China and the North China Plain. In additionAdditionally, the warming trends                                    |

|----|---------------------|
| 1  |                     |

593 of  $T_{s-max}$  and  $T_{a-max}$  became more statistically significant in the North China Plain and 594 South China (see Figs. S510).

To further prove thisclearly illustrate these changes, we selected two regions in China for further investigation: R1 primarily includes included the North China Plain and R2 primarily includes included the Loess Plateau, as shown in(see \_ Figs. 13Fig 9a). Although tThese regions share the same latitudes, However, the trend for  $SSR_{R_g}$ were showed substantially different (see Fig 9b), contrasting trends in the two regions (see Figs. 13b).

After adjusting for the impacts effect of SSRRs and precipitationP, the annual trends of for  $T_{s-max}$  and  $T_{a-max}$  in R1 increased by 0.304 and 0.118 °C 10yr-1, respectively, whereas while those in R2 just increased by only 0.025 and 0.016 °C 10yr-1, respectively. Therefore, following the adjustment, The the differences in the warming rates of  $T_{s-max}$  and  $T_{a-max}$  between R1 and R2 reduced-were significantly reduced after adjusting (see Figs. 13Fig 9d).

MeanwhileFollowing the adjustment – in R1, the seasonal and diurnal differences in the warming rates of  $T_{s-max}$  and  $T_{a-max}$  decreased – significantly\_decreased. After adjusting, in R1, tThe differences in warming rates between the warm seasons and cold seasons decreased by 68.7% for  $T_{s-max}$  and decreased by 50.8% for  $T_{a-max}$  after the **带格式的:** 下标

|----|---------------------|

|--------------------|--|--|

|---|---------------------|
|   |                     |

| 611 | adjustment . Additionally , the differences in the warming rates between $T_{s-max}$ and $T_{s-max}$ .                                   | _ | 【 带格式的: 字体: 倾斜
【 带格式的: 字体: 倾斜             |
|-----|--------------------------------------------------------------------------------------------------------------------------------------------------------|---|------------------------------------------------------------|
| 612 | min decreased by 93.4% and that between $T_{a-max}$ and $T_{a-min}$ decreased by 59.6% in R1. In                                                       | _ |  <li>【 带格式的: 字体: 倾斜</li> <li>【 带格式的: 字体: 倾斜</li>  |
| 613 | R2, the adjustment did not significantly change the seasonal and diurnal differences in                                                                |   |                                                            |
| 614 | temperatures. The seasonal and diurnal difference of temperatures in R2 had no                                                                         |   |                                                            |
| 615 | significant changes after adjusting. All in allOverall, the trends of for R1 and R2 became                                                             |   |                                                            |
| 616 | more consistent with each other after adjusting the for difference in $SSR_{R}$ and precipitation P between them (see Figs. 13 Fig. 9d). | _ |  <li>( 带格式的: 下标</li> <li>( 带格式的: 字体: 倾斜</li>      |
| 617 | 4. Conclusions and Discussion                                                                                                                          |   |                                                            |
| 618 | In China, despite the Although a general warming trends has been observed                                                                              |   |                                                            |
| 619 | throughout China, over the entire country, the regional warming trends showed                                                                          |   |                                                            |
| 620 | significant spatial and temporal heterogeneity. In this paperstudy, we analyzed the                                                                    |   |                                                            |
| 621 | spatial and temporal patterns of $T_s$ and $T_a$ from 1960 to 2003 and further analyzed and                                                            |   |                                                            |
| 622 | quantified the impact effects of $SSRR_{s}$ and precipitation P on these temperatures. The                                                             | _ |  <li>(帶格式的: 下标</li> <li>(帶格式的: 字体: 倾斜</li>        |
| 623 | main-primary results of the study are as follows.                                                                                                      |   |                                                            |
| 624 | The national mean warming rates from 1960 to 2003 of $T_{s-max}$ , $T_{s-min}$ , $T_{a-max}$ , and                                                     |   |                                                            |
| 625 | $T_{a-min}$ were 0.227 $\pm$ 0.091, 0.315 $\pm$ 0.058 $-$ °C 10yr -1 , 0.167 $\pm$ 0.068 $-$ °C 10yr -1 , and                    |   | 【 带格式的: 字体: 倾斜                                      |
| 626 | $0.356 \pm 0.057$ °C 10yr -1 , respectively. from 1960 to 2003. The he warming rates of $T_{s-1}$                                           |   | 【 带格式的: 字体: 倾斜                                      |
| 627 | min and $T_{a-min}$ were significantly greater than those of $T_{a-max}$ and $T_{a-max}$ (see Figs. 2 and                                              |   | 【 带格式的: 字体: 倾斜                                      |
| 628 | Figs. 3). Warming warming rates of $T_{s-max}$ and $T_{a-max}$ in South China and on-the North                                                         | _ |  <li>【 带格式的: 字体: 倾斜</li> <li>【 带格式的: 字体: 倾斜</li>  |

629 China Plain were significantly lower than those in the other regions (see Figs. 4)., and The the spatial

630 heterogeneity in the warm seasons was greater than that in the cold seasons.

During the study period, the SSRRs value\_decreased by  $-1.502\pm0.042$  W m-2 10yr-1 (95% confidence level)in China, with and higher dimming-diminishing rates were observed in South China and the North China Plain. Using a partial regression analysis, we found that SSRRs was the primary cause of the spatial patterns in the warming rates of Ts-max and Ta-max.

636 After adjusting for the effect of  $R_s$  and P, the warming rates of  $T_{s-max}$  and  $T_{a-max}$  in 637 South China and the North China Plain significantly increased and the regional 638 differences in warming rates in China clearly decreased (see Fig 8). After the 639 adjustments, the warming rates of Ts-max and Ta-max in the North China Plain increased 640 by 0.304 and 0.118 °C 10yr-1, respectively, whereas those on Loess Plateau increased 641 only by 0.025 and 0.016 °C 10yr-1, respectively. Therefore, the differences in warming 642 rates of Ts-max and Ta-max between the North China Plain and the Loess Plateau were 643 almost eliminated (see Fig 9d).

After adjusting for the effect of  $R_s$  and P, the warming trend of  $T_{s-max}$  increased by 0.146 °C 10yr-1 and that of  $T_{a-max}$  increased by 0.055 °C 10yr-1. In addition, the trends of  $T_{s-max}$  and  $T_{a-max}$  became 0.373±0.068 and 0.222±0.062 °C 10yr-1 respectively. 带格式的: 下标

 下标

 字体:
 倾斜

| 647 | Reduction in $R_s$ resulted in decreases in the warming rates of $T_{s-max}$ and $T_{a-max}$ by                                                                        |
|-----|------------------------------------------------------------------------------------------------------------------------------------------------------------------------|
| 648 | 0.139 °C 10yr -1 and 0.053 °C 10yr -1 , respectively, which accounted for 95.0% and 95.8%                                                        |
| 649 | of the total effect of $R_s$ and $P$ , respectively. For the seasonal contrast, the warming rates                                                                      |
| 650 | of $T_{s-max}$ and $T_{a-max}$ decreased by 45.0% and 17.2%, respectively. For the daily contrast,                                                                     |
| 651 | the warming rates of $T_s$ and $T_a$ decreased by 33.0% and 29.1%, respectively. After                                                                                 |
| 652 | adjusting for the impact of SSR and precipitation, the warming trend of T s-max increased                                                                   |
| 653 | by 0.146 °C 10yr -1 and that of T a-max -increased by 0.055 °C 10yr -1 . After adjustments,                                           |
| 654 | the trends of T s max , T s min , T a max , and T a min became 0.373 °C 10yr =1 , 0.315 °C 10yr -1 , |
| 655 | 0.222 °C 10yr -1 , and 0.356 °C 10yr -1 . The reduction of SSR resulted in the warming                                                           |
| 656 | rates of T s max and T a max decreasing by 0.139 °C 10yr -1 and 0.053 °C 10yr -1 , accounting                              |
| 657 | for 95.0% and 95.8%, respectively, of the total impact of SSR and precipitation.                                                                                       |
| 658 | In addition to SSRRs and precipitationP , temperatures 2 warming rates may be                                               |
| 659 | affected by many other factors, such as land cover and land use changes, ; that however                                                                                |
| 660 | those factors have not been discussed in this study due tobecause of lack of data; i.e.,                                                                               |
| 661 | land cover and land use (Liu et al., 2005; Zhang et al., 2016). After adjusting for the                                                                                |
| 662 | impact effect of changes in SSRR and precipitation P changes, the spatial differences                                                                    |
| 663 | in the warming trends clearly decreased; however, some certain regional differences                                                                                    |
| 664 | remained. The warming rate of $T_{s-max}$ in the Sichuan Basin remained significantly lower                                                                            |
| 665 | than that in other regions after adjusting for these impactseffects . In                                                                                 |

|----|---------------------|

| 666               | additionAdditionally, the differences north-south difference in the warming rates of $T_{s}$ .                                                                                                                                                              |
|-------------------|-------------------------------------------------------------------------------------------------------------------------------------------------------------------------------------------------------------------------------------------------------------|
| 667               | min and $T_{a-min}$ between the northern and southern areas were not cannot be explained by                                                                                                                                                                 |
| 668               | the impacts effects of SSRRs and precipitation P:- Further further study is                                                                                                                                                                                 |
| 669               | neededrequired.                                                                                                                                                                                                                                             |
|                   |                                                                                                                                                                                                                                                             |

[revised manuscript text omitted]
| 913                      |                                                                                                                                                                                                                                                                                                   |
| 914                      |                                                                                                                                                                                                                                                                                                   |
| 915                      |                                                                                                                                                                                                                                                                                                   |
| 916                      |                                                                                                                                                                                                                                                                                                   |
| 917                      |                                                                                                                                                                                                                                                                                                   |
| 918                      |                                                                                                                                                                                                                                                                                                   |
| 919                      |                                                                                                                                                                                                                                                                                                   |

---

## Referee Report (RR1)

"Contribution of surface solar radiation and precipitation to spatiotemporal patterns of surface and air temperature warming in China from 1960-2003" by Du et al.

After reading the manuscript, I have the following main comments

1) While the regional trend patterns of Ts, Ta, Rs and P and their relations are interesting, the authors did not give fully explanation of the possible reasons causing the formation of these patterns.

2) Unless I downloaded the wrong version, I found there are some mislabelling of Figures in the text. e.g., I can't find Fig S9g-i (line 383).

3) I noticed the authors were asked to justify the calculation of adjusted T, but I can't see the justification in the current version.
These parameters (Ta, Ts, Rs and P) interacts with each other, it is not easy to distinguish the casual relations. I am not sure I understand the actual implication of the adjusted T in section 3.3.

Minor comments

Please make Figure or Fig. consistent. eg Figure 1, but Fig. S1.

Line 22: "**stronger in northwest China**". Are you sure it is not the northeast China?

Line 22: "**and weaker in South China**". Is this before adjust or after adjust? Before adjust as shown in Fig. 4, the trend is negative in South China, not "weaker".

Line 23: Rs is redefined as SSR in Fig. S4. Make it consistent.

Line 25: Is Rs daily mean, monthly mean here? Same for Ts-max and T-a-max

Fig.S6: the caption is copied from the old version, the referenced Figs are not there anymore.

Line 28: "**were much higher than those in other regions**". It is hard to get this conclusion, since you don't have enough coverage over other areas.

Line 32: "**North China Plain and the Loess Plateau".** It will be easier to see if you mark these regions on the corresponding map.

Line 92: "**sever air pollution** "  severe air pollution

Line 155: "**Our study is the first to use *Ts* observation as a parameter for identifying regional climate change**."  I doubt it.

Line 196: "**reanalyzes**" , reanalyses?

Line 241: In Eq (1). Are Rs and P anomalies? Please make it clear.

Line 245: "**Fig. S3**". Fig. S4?

Fig. S4 & 5: Please explain the large difference of the sensitivity in Fig S5a and g.

Line 342: Fig S7? Please check it is the right Fig.

Line 376: Fig. S5? Please check.

Line 381-382: Fig S9 doesn't have g-i. Please check.

---

## Author Response (AR2)

**Manuscript**

"Contribution of Surface Solar Radiation and Precipitation to Spatiotemporal Patterns of Surface and Air Temperature Warming in China from 1960 to 2003" by Jizeng Du et al.

**Response to Reviewer # 1**

**1. Major**

**Comment:** While the regional trend patterns of $T_s$, $T_a$, $R_s$ and $P$ and their relations are interesting, the authors did not give fully explanation of the possible reasons causing the formation of these patterns.

**Reply:** Following the reviewers comments, we added two paragraphs to explain why the changes of $R_s$ and $P$ caused the pattern of $T_s$ and $T_a$ (Lines 120-144):

"It is well known that the diurnal cycles in $T_a$ and $T_s$ are primarily determined by the surface energy budget. After sunrise, the surface absorbs solar radiation, and the surface net radiation becomes positive and heats the surface first. As a result, the air above the surface becomes unstable. Surface net radiation can be partitioned into three parts: ground heat flux, sensible heat flux, and latent heat flux. Ground heat flux heats the surface and stores energy during the daytime, and this energy may be re-emitted at night. Sensible heat flux directly heats the air above the surface. Latent heat flux is the energy employed to vaporize water during the surface water evaporation and vegetation transpiration processes. How surface net radiation partitions into ground heat flux, sensible heat flux, and latent heat flux is determined by both surface and atmospheric conditions (Wang et al., 2010a, b; Wang and Dickinson, 2012), i.e., surface wetness. Daytime surface net radiation is primarily determined by $R_s$ (Wang and Liang, 2008) and precipitation or surface wetness control partition of surface net radiation into latent and sensible fluxes (Wang and Liang, 2008). Therefore, it is expected that changes in $R_s$ and $P$ play a key role in the variability of $T_s$ and $T_a$ (Wild, 2012; Manara et al., 2015; Hartmann et al., 1986)."

However, quantitative assessments of the impact of $R_s$ on $T_s$ and $T_a$ are still lack due the shortness of high quality of long-term estimates of $R_s$. In this study, we used sunshine duration derived $R_s$ (Wang, 2014; Wang et al., 2015) to quantitate the impact of $R_s$ on the spatial pattern of $T_a$ and $T_s$. To our knowledge, this study presents the first analysis of the relationship between $R_s$ (and $P$) and $T_a$ (and $T_s$) based on their spatiotemporal patterns and we further quantified the effect of variations of $R_s$ (and $P$) on $T_a$ (and $T_s$) in China for the period 1960-2003."

**Comment:** Unless I downloaded the wrong version, I found there are some mislabelling of Figures in the text. e.g., I can't find Fig S9g-i (line 383).

**Reply:** The reviewer has an incorrect version of the supplementary material. The supplementary material has been substantially revised during last revision. We double checked and make sure it is correct in this upload.

**Comment:** I noticed the authors were asked to justify the calculation of adjusted $T$, but I can't see the justification in the current version. These parameters ($T_a$, $T_s$, $R_s$ and $P$) interacts with each other, it is not easy to distinguish the casual relations. I am not sure I understand the actual implication of the adjusted $T$ in section 3.3.

**Reply:** We agree with the reviewer that the regression made in this paper does not provide any information on the casual relation. Following the reviewer's suggestion, we added two paragraphs to explain why the changes of $R_s$ and $P$ impacted the pattern of $T_s$ and $T_a$ (Lines 120-142). The reason is well known, however, the quantitative assessment on this aspect is still lacking because of the shortness of data. Due to continuous effort, we reconstructed long-term time series of surface solar radiation based on the sunshine duration observations. Furthermore, this study first found that the $T_s$ is most suitable parameter to study the issue, which has been ignored by the other researchers. We revised the equations and justifications based on the reviewers' suggestion during last round of review.

**2. Minor**

**Comment:** Line 54: Please make Figure or Fig. consistent. E g Figure 1, but Fig. S1.

**Reply:** We have revised all Figure X to Fig. X in new manuscript.

**Comment:** Line 22: "stronger in northwest China". Are you sure it is not the northeast China?

**Reply:** Thanks for your comments. We agree with you that the warming trend is not only stronger in "northwest China" but also in "northeast China". So we have corrected this sentence in Line 20-21: "the pattern was stronger in northwest and northeast China and weaker or negative in South China and the North China Plain."

**Comment:** Line 22: "and weaker in South China". Is this before adjust or after adjust? Before adjust as shown in Fig. 4, the trend is negative in South China, not "weaker".

**Reply:** This is warming trend before adjusting. We agree with your comments and have corrected this sentence in Line 20-21: "the pattern was stronger in northwest and northeast China and weaker or negative in South China and the North China Plain".

**Comment:** Line 23: Rs is redefined as SSR in Fig. S4. Make it consistent.

**Reply:** The reviewer has the incorrect version of the supplementary material. It is correct in the new version.

**Comment:** Fig.S6: the caption is copied from the old version; the referenced Figs are not there anymore.

**Reply:** The reviewer has the incorrect version of the supplementary material. It is correct in the new version.

**Comment:** Line 28: "were much higher than those in other regions". It is hard to get this conclusion, since you don't have enough coverage over other areas.

**Reply:** Thanks for your comments. We revised this sentence to (Lines 25-27):

"More importantly, the decreasing rates in South China and the North China Plain were stronger than those in other parts of China."

**Comment:** Line 32: "North China Plain and the Loess Plateau". It will be easier to see if you mark these regions on the corresponding map.

**Reply:** Thanks for your valuable comments. We have marked these regions on the Fig. 9a and introduced them in Line 443-445: "To clearly illustrate these changes, we selected two regions in China for further investigation: R1 primarily included the North China Plain and R2 primarily included the Loess Plateau (see Fig. 9a)."

**Comment:** Line 92: "sever air pollution "severe air pollution

**Reply:** We have corrected it in Line 89.

**Comment:** Line 155: "Our study is the first to use Ts observation as a parameter for identifying regional climate change." I doubt it.

**Reply:** We have deleted this sentence in new manuscript.

**Comment:** Line 196: "reanalyzes", reanalyses?

**Reply:** We have revised it in new version.

**Comment:** Line 241: In Eq (1). Are $R_s$ and $P$ anomalies? Please make it clear.

**Reply:** Following your comment, we have added a statement that in Line 256-258: "$R_s$ and $P$ represents the monthly anomalies of surface solar radiation and precipitation respectively".

**Comment:** Line 245: "Fig. S3". Fig. S4?

**Reply:** The reviewer has the incorrect version of the supplementary material. It is correct for the new version.

**Comment:** Fig. S4 & 5: Please explain the large difference of the sensitivity in Fig S5a and g.

**Reply:** We have explained the difference between the sensitivities of $T_s$ and $T_a$ to $R_s$ in Line 339-343: "During the day, $T_s$ is directly determined by the land surface energy balance, i.e., the incoming energy (including $R_s$) and atmospheric longwave radiation (Wang and Dickinson, 2013a), and it is partitioned into latent and sensible heat fluxes (Zhou and Wang, 2016). Although $T_a$ is dependent on the land-atmosphere sensible heat flux, it is also affected by local and/or large-scale circulation."

**Comment:** Line 342: Fig S7? Please check it is the right Fig.

**Reply:** The reviewer have the incorrect version of the supplementary material. It is correct for the new version.

**Comment:** Line 376: Fig. S5? Please check.

**Reply:** The reviewer has the incorrect version of the supplementary material. It is correct for the new version.

**Comment:** Line 381-382: Fig S9 doesn't have g-i. Please check.

**Reply:** The reviewer has the incorrect version of the supplementary material. It is correct for the new version.

**Response to Reviewer # 2**

**1. General Comments**

**Comment:** There is one conceptual issue: the use in L26 and L451 of "caused" for results from the regression analysis between $T$ and $R_s$. It is not considered acceptable from a regression analysis of A and B to say that A caused B, even though in a simplified surface-boundary layer model, $R_s$ is the primary physical driver of $T_{max}$. In the real world, other factors (L468) that you have not measured may be involved, and some others such as the outgoing LW are themselves coupled to $T_{max}$. Your point is that if you use the slope of the partial regression of $T_{max}$ on $R_s$ to remove the spatial variation of $R_s$, the $T_{max}$ trends became spatially less heterogeneous. Find other wording that allows you to make your point.

**Reply:** We agree with the reviewer that the regression made in this paper does not provide any information on the casual relation. Following the reviewer's suggestion, we added two paragraphs to explain why the changes of $R_s$ and $P$ caused the pattern of $T_s$ and $T_a$ (Lines 120-142). The reason is well known, however, the quantitative assessment on this aspect is still lacking because of the shortness of data. Due to the continuous effort, we reconstructed long-term time series of surface solar radiation based on the sunshine duration observations. Furthermore, this study first found that the $T_s$ is most suitable parameter to study the issues, which has been ignored by the other researchers. We revised the equations and justifications based on the reviewers' suggestion during last round of review.

**2. Minor**

**Comment:** L92 severe air pollution

**Reply:** We have corrected it as your comments in new version.

**Comment:** The notation $R_s/P$ for $R_s$ and $P$ is not conventional as / is normally used for 'divided by'. Also Ts-max/ Ta-max

**Reply:** We have replaced all "$R_s/P$" to "$R_s$ and $P$", same to "$T_{s\text{-}max}/ T_{a\text{-}max}$".

**Comment:** L278 increased by 0.356±0.0057 °C 0.356±0.057 °C?

**Reply:** We have revised it in new version.

**Comment:** L307 This dependence has been successfully attributed to amplified dynamics. This comment needs clarification

**Reply:** Thanks for your comments. We have changed the expression and make it clearer. "Related studies suggested that this dependence was strongly associated with the mode variability in large-scale circulation, such as a negative trend in the North Atlantic Oscillation during this period (Wallace et al., 2012; Ding et al., 2014)." (Line 324-327).

**Comment:** L789 which did not include the effect of the $R_s$ variations. Are you sure you mean this? Or 'after removing the effect of the $R_s$ variations' (and $P$?)

**Reply:** Thanks for your comments. We have corrected those sentences: "(c) Annual, warm, and cold seasonal scale trends calculated based on the data before adjusting the effect of $R_s$ and $P$. (d) Annual, warm, and cold seasonal scale trends calculated based on the data after adjusting the effect of $R_s$ and $P$." (Line 815-817).

**Comment:** L451 'primary cause of" (and L26 of abstract). Find other wording since regression shows correlation and association, but not causes. So it is incorrect to say 'primary cause'. See also L468-469 where other factors, not measured are mentioned. Reword L326 as well.

**Reply:** Please see also our response to your general comments. We have revised the wording as your suggestion. "
[revised manuscript text omitted]